# Accuracy of AI-based binary classification for detecting malocclusion in the mixed dentition stage

Kengo Oka◉*, Saki Uemura, Satoru Morishita, Yukio Yamamoto, Kei Kurita◉

Advanced Oral Health Science Research Laboratories, Research and Development Headquarters, LION Corporation, Edogawa-ku, Tokyo, Japan

* k-oka@lion.co.jp

## Abstract

### Background

Malocclusion is a common anomaly and is frequently observed in children and adults. Early detection and treatment of malocclusion is necessary to prevent and minimize complications. Therefore, developing a tool to check dentition at an early stage and motivate patients themselves to visit the dentist is required.

### Objective

This study aimed to examine the feasibility of building an AI model that can detect malocclusion in children during the mixed dentition stage.

### Methods

This study was conducted as a feasibility study using cross-sectional data. Subjects were recruited from panelists registered with Macromill, Inc. (approximately 1.3 million registered in 2021). A total of 519 elementary school children (275 boys and 244 girls in Grades 3–6) were included in this study. Questionnaire data and tooth alignment images of the children were collected. The dataset was created, and AI-based binary classification models for malocclusion were developed using an automated machine learning platform (DataRobot) to construct three algorithms for determining malocclusion (deep bite, maxillary protrusion, and crowding). Using a test dataset, the model's performance was assessed through sensitivity, specificity, accuracy, precision, F1 score, receiver operating characteristic (ROC) curves, and area under the ROC curves (AUC).

### Results

Three dental images were used for all model building, and questionnaire data used all four questions about oral habits (Q1: mouth open during the day, Q2: sleep with

**Data availability statement:** Data cannot be shared publicly because of a lack of such description in the study protocol. For items not listed in the research protocol, data sharing is restricted by Instituitonal Review Board of LION Corporation. Data are available from the Instituitonal Review Board of LION Corporation for researchers who meet the criteria for access to confidential data. (Contact via Kazuo Mukasa (Instituitonal Review Board member of LION Corporation), E-mail: mukkaz@lion.co.jp).

**Funding:** This work was funded by Lion Corporation (https://www.lion.co.jp/en/).

**Competing interests:** I have read the journal's policy and the authors of this manuscript have the following competing interests: [The authors of this manuscript employed by LION Corporation. Recruitment of participants was performed by Macromill Inc. (Tokyo, Japan) under consignment from Lion Corporation.].

mouth open, Q3: have difficulty eating hard foods, Q4: prefer soft foods) for the deep bite classification model, Q1 and Q3 for the maxillary protrusion classification model, and Q1 and Q4 for the crowding classification model. The maxillary protrusion and crowding classification models showed moderate accuracy (AUC > 0.70), and the deep bite classification model showed high accuracy (AUC > 0.90). The permutation importance showed that dental image was the highest contributing factor in each model. Furthermore, while questionnaire data on oral habits were not an important factor in determining deep bite, these questionnaire data were an important factor in determining maxillary protrusion and crowding. Also, statistical analysis of the association between malocclusion and these oral habits revealed a significant association between maxillary protrusion or crowding and the presence or absence of oral habits.

## Conclusion

For the detection of malocclusion in mixed dentition, AI-based binary classification models are a promising approach as a screening tool.

## Introduction

Malocclusion is defined as a misalignment of teeth in the two dental arches or an incorrect tooth-to-tooth relationship. A systematic review by Alhammadi et al. revealed a high global prevalence of malocclusion [1]. According to the study, during the mixed dentition stage, the average prevalence of malocclusion in Mongoloids was 66.75% for Class I, 22.1% for Class II, and 10.95% for Class III. In Caucasians, the average prevalence was 70.39% for Class I, 25.91% for Class II, and 3.53% for Class III. The results of the above studies indicate that the prevalence of malocclusion in classes I and II is quite high worldwide. Malocclusion not only affects masticatory function [2] but also adversely affects psychological aspects, such as confidence in appearance, and is a secondary cause of dental caries [3]. For this reason, it is recommended that patients with malocclusions be treated as early as possible. Despite the large proportion of people with malocclusion, the proportion of people receiving orthodontic treatment in Japan is low. This may be because patients themselves are unaware that the degree of malocclusion is severe enough to require treatment. Castellote et al. found that the perception of malocclusion varies among individuals, patients, and practitioners [4]. They examined the perception of dental aesthetics among orthodontists, general dentists, and the general population by using frontal intraoral photographs cases classified by the Dental Aesthetic Index (DAI). The results revealed significant differences in Index of Orthodontic Treatment Need-Aesthetic Component (IOTN-AC) scores, with orthodontists giving the most severe scores. Another study compared the self-perception of dental aesthetics among 173 Japanese university students with dentists' assessments using the IOTN-AC scale [5]. The comparison showed a large discrepancy: Of the 33 students judged by dentists to need treatment, only 3 (9.1%) believed they need it. These studies suggest that some patients may not be aware that their malocclusion is severe enough to require treatment.

A previous study of nine OECD (Organization for Economic Cooperation and Development) member countries with low DMFT (Decayed, Missing, and Filled Teeth) indices found that Japan's regular dental visit rate at age 12 was 44%, compared with 85–99% in other countries, indicating that the rate of regular dental visits in Japan is low [6]. This indicates that guardians and children in Japan have few opportunities to receive feedback from their doctors on their children's teeth alignment, thus missing the opportunity to receive treatment at the appropriate time. Therefore, there is a need for a checking tool that will motivate people to visit the dentist and become aware of their teeth alignment. Malocclusion is diagnosed through detailed examinations, such as X-rays. However, in large-scale health checkups and epidemiological research, methods such as DAI and IOTN are used to screen for malocclusion. While these indices are useful for quantifying the need for orthodontic treatment, they are time-consuming and may result in inter-rater variability depending on the evaluator. Therefore, there is a need for a screening tool for malocclusion that is easier to use and has less inter-rater error.

Recently, AI classification models using images have been studied, and an algorithm for classifying oral conditions, such as periodontal disease and caries, has been reported [7,8]. Furthermore, Ha *et al.* applied the algorithm of the CNN (convolutional neural network), which is often used in object detection, to dental panoramic images, and they reported that an object detection CNN model based on YOLOv3 was able to detect the presence of a single impacted mesiodens on panoramic radiographs of primary, mixed, and permanent [9]. Based on these reports, it is anticipated that we can develop algorithms for a simple oral health evaluation by using oral images and machine learning (e.g., CNN). This would not only provide patients with appropriate feedback regardless of their location but also serve as a screening tool to promote clinics visits and enable the detection of oral diseases in places where detailed examinations are difficult, such as at home or during health checkups.

However, in addition to the fact that there is no reports on AI models for determining malocclusion during the mixed dentition period, in order to make the prediction algorithm available to the general public at home or health checkup venues, it is necessary to construct an algorithm using dental images that can be taken at home, rather than panoramic photographs taken at dental clinics. Furthermore, intraoral photographs taken at dental clinics are often taken for the purpose of diagnosing malocclusion, which may result in a biased dataset with a high proportion of individuals with malocclusion. To build an AI model that detects malocclusion, it is necessary to also collect images that do not show malocclusion abnormalities, and data diversity is required.

Therefore, this study aimed to examine the feasibility of building an AI model that can check tooth alignment in children regardless of location. In this study, approximately 1,500 images of children's teeth captured with smartphones were collected online, and a demonstration experiment to build an algorithm to determine various tooth alignment conditions was conducted.

## Materials and methods

### Study participants

The structure of the present manuscript followed the guidelines for Developing and Reporting Machine Learning Predictive Models in Biomedical Research [10]. In this study, subjects participated in the study from May 28, 2021 to July 31, 2021. This study was conducted as a feasibility study using cross-sectional data and included elementary school children (grades 3–6). Children who reported a history of trismus, underwent orthodontic treatment or were undergoing orthodontic treatment, and had pain when opening their mouth (when talking or eating) were excluded from this study. Furthermore, this study included participants who had smartphones with Android 6.0 or higher or iOS 11.0 or higher. Participant eligibility was assessed using questionnaires completed by the guardians on the web page. Finally, a total of 520 Japanese children were included in this study (Fig 1). The study protocol was approved by the Institutional Review Board of Lion Corporation (Clinical Trial Registration number 351) and conducted in accordance with the Declaration of Helsinki and the Japanese Ethical Guidelines for Epidemiological Research. Recruitment, selection, and consent of subjects were

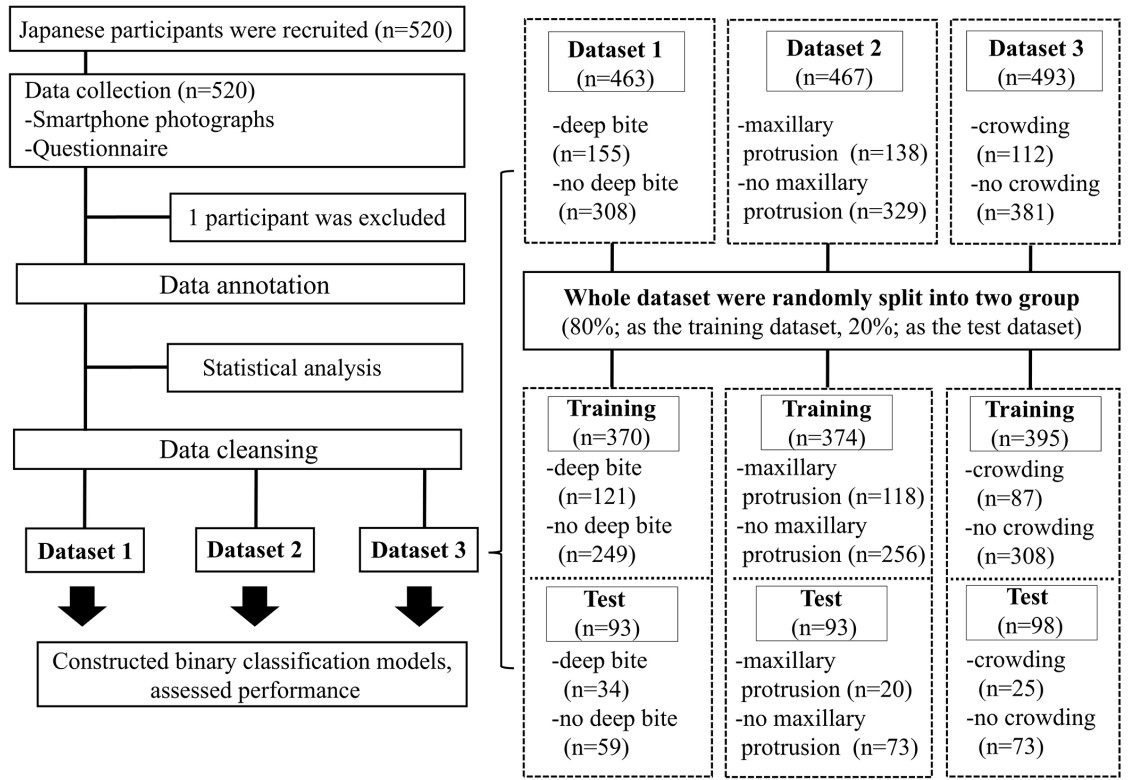

**Fig 1. Flowchart of study participants and design.** A total of 520 Japanese children were recruited. They were assessed for eligibility. Finally, 520 children were registered as participants. During this study, 1 participant was excluded (not meeting the inclusion criteria). After data cleansing, three datasets were created to build each model to determine the presence of deep bite, maxillary protrusion, and crowding, respectively.

performed by Macromill Inc. (Tokyo, Japan) under consignment from LION Corporation. Participants in this study were recruited from among the 1.3 million panelists members registered with Macromill, Inc. as test participants at the time of the 2021 survey (Macromill's registered panelists represent a diverse group in terms of residence and household income that does not differ greatly from the general population). Macromill Inc. recruited subjects, explained the content of the study and other information on the Web screen, obtained consent for participation in the study based on the subject's (guardian's) free will on the Web. When obtaining consent, a consent form was displayed on the web screen and buttons were provided for subjects to indicate whether they agreed or disagreed. For children participating in the study, Macromill, Inc. selected the child's parent or guardian as a proxy consentor, and obtained the proxy's free and voluntary consent via the Web. In addition, the proxy consulter himself/herself fully explained the contents of the survey to the child and obtained the child's free and voluntary consent before conducting the survey. Consent was obtained by placing a response button on the website. Parents or guardians were asked to indicate whether they had obtained consent from their child by pressing the button. The date and time of these web-based responses were recorded as the date and time of the consent response. Informed consent was obtained from all participants involved in this study.

## Data collection

All surveys were conducted online. Survey data were collected anonymously using a web-based survey website system owned by the research firm (Macromill Inc). When guardians first accessed the survey site, they were presented with a policy regarding the use of collected data and the protection of their personal information. Guardians followed the

instructions and completed a questionnaire on their children's oral habits (Table 1). Tooth alignment images obtained by smartphone were collected using the following method: (1) Guardians installed an application for capturing tooth alignment images following the research company's instructions. (2) After installing the application, the guardians were instructed to access a designated website and watch a lecture video on how to capture tooth alignment images. (3) After watching the video, the guardians opened the application installed on a smartphone and captured three images (front, left side, and right side) of the child's tooth alignment according to the guide frame on the application (Fig 2). In addition, before the shooting guide frame is displayed, the following instructions were displayed on the application screen to standardize the shooting conditions. The application interface is designed to require users to press the confirm button after reading the instruction. The instructions for shooting conditions are as follows. (a) take the photo in a well-lit place, (b) take the photo with the height of the child's face and the height of the smartphone camera aligned, (c) take the photo with the back teeth firmly clenched, and (d) take the photo with the lips open so that the teeth are visible). (4) If out-of-focus or other unclear images were obtained, guardians re-took the images until appropriate images were obtained. (5) The guardians uploaded tooth alignment images taken to a web page designated by the research company. Finally, questionnaire data and tooth alignment images (front, left side, and right sides) were collected from the 520 participants (Fig 3). Upon review of the image data, one participant was excluded from the study because the image data showed dentition that appeared to be that of an adult, not a child. Finally, this study included 519 elementary school children (275 boys and 244 girls in Grades 3($n = 157$), 4 ($n = 172$), 5 ($n = 110$), and 6 ($n = 80$)). In addition, the number of dental images collected from the 519 participants is as follows: 519 frontal dental images, 513 right side dental images, and 512 left side dental images, for a total of 1,544 images. There were no missing values in the questionnaire data, and responses were obtained from all 519 participants.

## Data analysis by machine learning

AI-based binary classification models for malocclusion were developed using the DataRobot, an automated machine learning platform. The DataRobot enterprise AI platform (DataRobot, Tokyo, Japan) was used to construct the algorithm

**Table 1. Each variable for the dataset.**

| Objective variables (binary data) | The presence of malocclusion (deep bite, maxillary protrusion, and crowding) | |
|---|---|---|
| Explanatory variables | Image | Dentition image (front) |
| | | Dentition image (left side) |
| | | Dentition image (right side) |
| | Questionnaire (binary data) | Q1: Is your child's mouth often open during the day? |
| | | Q2: Does your child sleep with their mouth open? |
| | | Q3: Does your child have difficulty eating hard foods? |
| | | Q4: Does your child often prefer soft foods? |

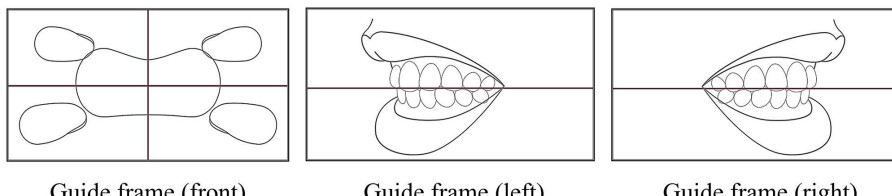

Guide frame (front)        Guide frame (left)        Guide frame (right)

**Fig 2. Guide frame that appears on the smartphone application.** The guardians captured three images (front, left side, right side) of the child's tooth alignment according to the guide frame on the smartphone application.

                                                                              

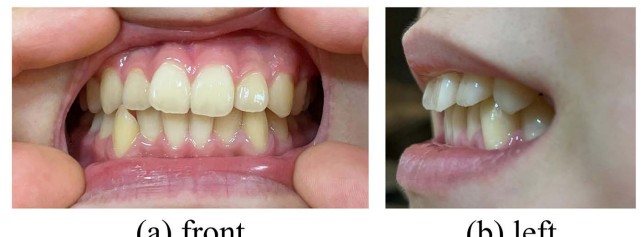 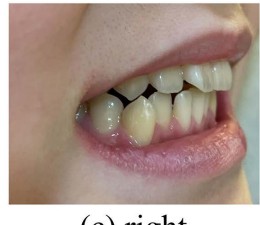

| (a) front | (b) left | (c) right |

**Fig 3. Representative photographic image.** Representative photographic images of the frontal view (a), left side (b), and right side (c) of the child's tooth alignment.

for classifying malocclusion. The AI platform provides a method for creating robust ensemble models using several machine learning algorithms [11,12]. The algorithm was constructed using the following steps. In the variables shown in Table 1, missing information was not supplemented with representative values or other substitute values, and was treated as missing values when constructing the algorithm. Note that due to the tool's specifications, DataRobot prioritizes higher accuracy over simpler model structures. Therefore, this study did not evaluate structural similarity between models and instead focused on models that demonstrated superior accuracy.

**Data annotation.** The dentition and occlusal status of children were comprehensively determined for the presence of each type of malocclusion from smartphone dental images based on the School Dentist Guidelines of the Japan Association of School Dentists and previous study [13]. The annotation criteria for each malocclusion type (deep bite, maxillary protrusion, crowding) used in constructing the three algorithms in this study are as follows: Deep bite: more than half of clinical crown in the lower front teeth are covered by the upper front teeth, crowding: adjacent teeth overlap by more than 1/4 of the width of the crown, maxillary protrusion: overjet is large (approximately 4 mm or more), indicating protrusion of the upper front teeth. When annotating the dental images collected, unclear images were excluded. Based on the above criteria, three researchers annotated the data according to the following procedure. (1) The three researchers simultaneously reviewed reference case photos corresponding to each type of malocclusion (deep bite, maxillary protrusion, and crowding) and calibrated their judgments. (2) Based on each dental image dataset, each researcher independently annotated whether each subject had the corresponding type of malocclusion. (3) The three researchers simultaneously reviewed the annotation data assigned to each of them, and they adopted the value for cases where all three researchers made the same judgment. (4) When the three researchers had different opinions, they reviewed the reference case photos again and re-calibrated their judgments. (5) For subjects with differing judgments, each researcher each re-annotated the data. (6) The three researchers simultaneously reviewed the re-evaluated annotation data. Identical judgments were adopted, and cases with remaining discrepancies were deemed unevaluable and excluded from the dataset. The final breakdown of the dataset regarding the presence of malocclusion after data annotation is as follows: Dataset for deep bite (n = 508): deep bite (n = 166)/ no deep bite (n = 342), dataset for maxillary protrusion (n = 515): maxillary protrusion (n = 147)/ no maxillary protrusion (n = 368), dataset for crowding (n = 518): crowding (n = 123)/ no crowding (n = 395).

**Data cleansing for machine learning.** To construct a classification model using deep learning, the annotated data were further cleaned. Subjects for whom at least one of the three dental images were clearly not taken in centric occlusion (e.g., cases where gaps were clearly visible, such as the tongue being visible through space, or where the occlusal relationship did not match between images, etc.) were excluded from the training dataset used to develop the classification models for deep bite and maxillary protrusion. Furthermore, data from subjects whose mandibular anterior teeth were inadequately captured in any of the three images due to deep bite or other reasons were excluded from the training dataset for the dental crowding classification model. The final breakdown of the dataset regarding the presence of

malocclusion after data cleansing is as follows: Dataset for deep bite (n = 463): deep bite (n = 155)/ no deep bite (n = 308), dataset for maxillary protrusion (n = 467): maxillary protrusion (n = 138)/ no maxillary protrusion (n = 329), dataset for crowding (n = 493): crowding (n = 112)/ no crowding (n = 381).

**Dataset.** The dataset was created using binary data on the applicability of deep bite, maxillary protrusion, and crowding as the objective variables and oral habit information and three dental images as the explanatory variables to construct three algorithms for determining malocclusion (deep bite, maxillary protrusion, and crowding). Table 1 shows the dataset items. The categorical variable, the presence or absence of the corresponding dentition type, was quantified as 1 for "applicable" and 0 for "not applicable," and the questionnaire response results were quantified as 1 for "yes" and 0 for "no" and used in the algorithm. Four questions on the presence or absence of the habit of open mouth and daily eating habits related to hard foods, which observational studies have suggested are associated with malocclusion [14–20], were selected for inclusion in the dataset (Table 1). Three datasets were created to build each model to determine the presence of deep bite, maxillary protrusion, and crowding, respectively (Fig 1). After creating each dataset, the whole dataset was randomly divided into two groups. Of the total data, 80% was randomly split as a training dataset for algorithm construction and the remaining 20% as a validation dataset.

**Modeling and validation.** Using the training dataset, three algorithms were constructed to determine the presence of deep bite, maxillary protrusion, and crowding. Models were created using the DataRobot. It was used to create a number of models, including "blender models" obtained using multiple machine learning algorithms [11]. The following modeling was performed by Datarobot on autopilot. Questionnaire data and image data were automatically separated from the dataset, and feature extraction from the image data was performed by a pre-trained deep learning model of SqueezeNet, a type of CNN that had been pre-trained. Next, Datarobot used the binary data from questionnaire data and the extracted image features as explanatory variables to create a classification model for the presence or absence of deep bite using the Elastic Net Classifier. In the maxillary protrusion classification model, before building the model using all data sets, Datatobot only used the feature values extracted by SqueezeNet, and calculated the predicted value for the presence or absence of maxillary protrusion using the Elastic Net Classifier. Then, Datarobot used the predicted value itself as a new feature value, integrated it with the binary data from the questionnaire data, and constructed an algorithm for judging maxillary protrusion using the Elastic Net Classifier. In the crowding classification model, before building the model using all data sets, Datarobot used only the image features extracted by SqueezeNet as explanatory variables and calculated predictions for the presence or absence of crowding using the Elastic Net Classifier. Datatobot adopted the above prediction values themselves as new features. By feeding the binary data from the questionnaire and the prediction values into the SVM Classifier model as explanatory variables, we constructed a crowding classification model.

All models were validated by fivefold cross-validation using the area under the receiver operating characteristic (ROC) curve (AUC) as the evaluation measure. Cross-validation scores were calculated by taking the average of the AUCs of the five validation folds. In terms of model selection, among the multiple models built, the model with the highest AUC value in five cross-validations was selected as the binary classification model. We considered inappropriate models and didn't select models that only focus on other areas such as fingers instead of the dental area in the Activation map even if the AUC value is high.

**Permutation importance (PI).** The relative importance of a parameter in the models was evaluated using the PI [21]. PI measures how worse the model's performance would be if DataRobot made predictions after randomly shuffling the elements in a given column (without changing the other columns). DataRobot calculated the importance of the most important explanatory factor as 100% and the importance of each other factor as relative values.

**Activation map.** The activation map (the area of the image focused in the constructed decision model) was created using the analysis features provided by Datarobot. DataRobot uses the Grad-CAM (Gradient-weighted Class Activation Map) technology to create the activation map [22]. The activation maps are overlaid on the images to show which image areas drive the model prediction decisions. Using this approach, we visualized where the AI model focuses on when

making judgments. To build an explainable AI model, heatmap plots were used to differentiate colors over the input image to show areas that had a significant impact on the AI model's decisions.

## Evaluation models

The validation dataset was used to evaluate the generalization performance of new data in the constructed model. The performance of the constructed models was assessed in terms of sensitivity, specificity, accuracy, precision, and F1 score. The values for the performance metrics were calculated in accordance with the following formulas:

$$Sensitivity = TP/(TP + FN)$$

$$Specificity = TN/(FP + TN)$$

$$Accuracy = (TP + TN)/(TP + FP + FN + TN)$$

$$Precision = TP \ / \ (TP + FP)$$

$$F1 \ score = 2 \ (Precision \times Sensitivity) \ / \ (Precision + Sensitivity)$$

where TP indicates true positive, TN indicates true negative, FP indicates false positive, and FN indicates false negative. Additionally, all models were evaluated using the AUC of the ROC curve as an evaluation measure.

## Comparison with logistics regression model

In order to compare the performance of the constructed machine learning algorithm with the classic logistic regression model, we have constructed a model using binary logistic regression analysis based on questionnaire data alone to determine the presence or absence of malocclusion, which was performed on Datarobot. Note that all four questionnaire data sets shown in Table 1 were used as explanatory variables to construct these logistic regression models. Furthermore, to enable comparison with machine learning models and their accuracy, this analysis employed the same dataset used for the machine learning models. As with the method described above, we divided the entire dataset into two random groups to build a logistic regression model. We used 80% of the data as a training dataset for model construction and the remaining 20% as a validation dataset. We then used the validation dataset to calculate the performance metrics described above and evaluate the model's performance.

## Statistical analysis

To test whether there was a relationship between the oral habits data used to construct the algorithm and the presence of malocclusion, the relationship between malocclusion and oral habits was analyzed using the chi-squared test. As shown in Fig 1, this analysis was performed using the dataset after data annotation (prior to data cleansing for machine learning model construction). Binary data on the applicability of each malocclusion (deep bite, maxillary protrusion, and crowding) were used as the objective variable, and oral habit information was used as the explanatory variable for statistical analysis. All data analyses were performed using JMP version 15.2.0 (SAS Institute Inc., Cary, NC, USA). A P-value <0.05 was considered statistically significant. Odds ratios (OR) and 95% confidence intervals were calculated using the analysis software.

## Results

A total of 519 children (275 boys and 244 girls) were included in this study. Of the 519 children, 157 were in Grade 3, 172 in Grade 4, 110 in Grade 5, and 80 in Grade 6. All measurements were collected. The following is a breakdown of the residential areas of the 519 participants recruited in this study (Table 2).

Of the 519 participants in this study, the breakdown of household annual income for the total of 447 participants who answered the household income question is shown below (Table 3).

The above percentages for annual household income show no much difference from the figures from the national census conducted by Japanese government agency for the entire country [23].

### Explicability results of the deep bite classification model

The frontal image (PI: 100%) was the most important explanatory variable in the model, followed by the right side image (PI: 31.7%) and left side image (PI: 23.9%) as important factors (Fig 4a). The PI for the questionnaire data was < 1%, indicating that it was not an important factor in predicting deep bite. The activation map showed that the area of focus in this model was near the bite of the anterior teeth (Fig 4b). The AUC score of this model was 0.8109, indicating moderate accuracy in the 5-fold cross-validation.

### Explicability results of the maxillary protrusion classification model

The frontal image (PI: 100%) was the most important explanatory variable in the model, followed by the left side image (PI: 91.4%), right side image (PI: 64.7%), questionnaire data on habits related to incompetent lip seal (Question 1) (PI: 19.0%), and questionnaire data on the habit of eating hard foods (Question 3) (PI: 4.21%) (Fig 5a). The activation map

**Table 2. Breakdown of participants by residential area.**

| Residential area | Number of participants (*n*) | Percentage of participants (%) |
|---|---|---|
| Hokkaido | 23 | 4.4 |
| Tohoku | 25 | 4.8 |
| Kanto | 167 | 32.2 |
| Chubu | 99 | 19.1 |
| Kinki | 104 | 20.0 |
| Chugoku | 31 | 6.0 |
| Shikoku | 15 | 2.9 |
| Kyusyu | 55 | 10.6 |

**Table 3. Breakdown of participants by annual household income.**

| Annual household income (yen) | Number of participants (*n*) | Percentage of participants (%) |
|---|---|---|
| under than 2 million | 15 | 3.4 |
| 2 to under 4 million | 68 | 15.2 |
| 4 to under 6 million | 148 | 33.1 |
| 6 to under 8 million | 116 | 26.0 |
| 8 to under 10 million | 59 | 13.2 |
| 10 to under 12 million | 27 | 6.0 |
| 12 to under 15 million | 9 | 2.0 |
| 15 to under 20 million | 4 | 0.9 |
| 20 or more million yen | 1 | 0.2 |

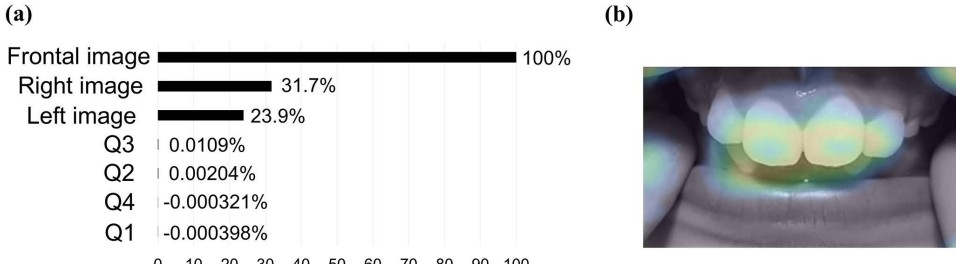

**Fig 4. Explicability results of the deep bite classification model.** (a) PI. (b) Representative photographic images of activation maps (frontal image). Red areas indicate areas of particular focus for the AI model to make decisions. The model was built using the Elastic-Net Classifier (L2/ Binomial Deviance) algorithm.

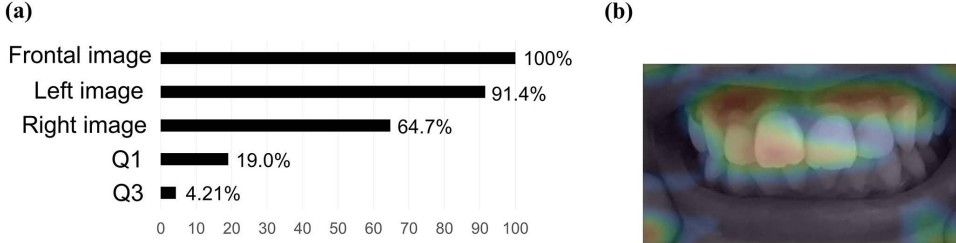

**Fig 5. Explicability results of the maxillary protrusion classification model.** (a) PI. (b) Representative photographic images of activation maps (frontal and left side images). Red areas indicate areas of particular focus for the AI model to make decisions. The model was built using the Elastic-Net Classifier (mixing alpha = 0.5/ Binomial Deviance) algorithm.

showed that the area of focus in this model was the maxillary anterior region (Fig 5b). The AUC score of this model was 0.8408, indicating moderate accuracy in the 5-fold cross-validation.

## Explicability results of the crowding classification model

The frontal image (PI: 100%) was the most important explanatory variable in the model, followed by the right side image (PI: 2.32%) and questionnaire data on the habit of eating hard foods (Question 4) (PI: 2.01%) (Fig 6a). The PI for the other items was < 1% or was not adopted by the algorithm, indicating that they were not important factors in predicting crowding.

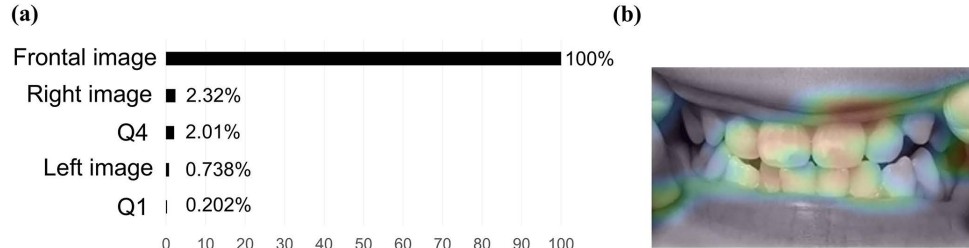

**Fig 6. Explicability results of the crowding classification model.** (a) PI. (b) Representative photographic images of activation maps (frontal image). Red areas indicate areas of particular focus for the AI model to make decisions. The Nystroem Kernel SVM Classifier algorithm was employed to build the model.

The activation map showed that the area of focus in this model was the entire tooth, not a specific spot (Fig 6b). The AUC score of this model was 0.7440, indicating moderate accuracy in the 5-fold cross-validation.

### Performance metrics for machine learning models on new data

The three machine learning models showed high performance, with an AUC of 0.70 or higher, which is an overall accuracy metric (Table 4). The deep bite model maintained high performance in detecting deep bite, with an AUC of 0.92, sensitivity of 91.2%, specificity of 81.4%. The maxillary protrusion model showed moderate accuracy, with an AUC of 0.74, sensitivity of 70.0%, specificity of 71.2%. The crowding model also showed moderate accuracy, with an AUC of 0.73, sensitivity of 80.0%, specificity of 61.6%. The other metric were shown in Table 4.

### Performance metrics for logistic regression models on new data

The three logistic regression models showed low performance with an AUC of less than 0.70 (Table 5). The deep bite model showed low performance, with an AUC of 0.52, sensitivity of 100.0%, specificity of 0.0%. The maxillary protrusion model also showed low accuracy, with an AUC of 0.55, sensitivity of 100.0%, specificity of 4.1%. The crowding model also showed low accuracy, with an AUC of 0.62, sensitivity of 48.0%, specificity of 75.3%. The other metric were shown in Table 5.

### Association between malocclusion and oral habits

Statistical analysis revealed no significant association between any of the questions and the presence or absence of deep bite (Table 6). However, oral habits related to incompetent lip seal (Questions 1 and 2) recorded significantly higher frequency in people with maxillary protrusion than in people with no maxillary protrusion (P<0.001 and P<0.05, OR = 2.16 and 1.60, respectively) (Table 7). Incompetent lip seal was suggested to be a risk factor for maxillary protrusion. Furthermore, oral habits related to avoiding hard foods (Questions 3 and 4) recorded significantly higher frequency in people with crowding than in people with no crowding (P<0.01, OR = 1.93 and 1.83, respectively) (Table 8). The result showed that oral habits related to avoiding hard foods to be a risk factor for crowding.

**Table 4. Performance metrics of the machine learning models on the test dataset.**

| Class | Sensitivity (%) | Specificity (%) | Accuracy (%) | Precision (%) | F1 score | AUC |
|---|---|---|---|---|---|---|
| Deep bite | 91.2 | 81.4 | 85.0 | 73.8 | 0.8158 | 0.92 |
| Maxillary protrusion | 70.0 | 71.2 | 71.0 | 40.0 | 0.5091 | 0.74 |
| Crowding | 80.0 | 61.6 | 66.3 | 41.7 | 0.5479 | 0.73 |

AUC: area under the receiver operating characteristic curve.

**Table 5. Performance metrics of the logistic regression models on the test dataset.**

| Class | Sensitivity (%) | Specificity (%) | Accuracy (%) | Precision (%) | F1 score | AUC |
|---|---|---|---|---|---|---|
| Deep bite | 100.0 | 0.0 | 36.6 | 36.6 | 0.5354 | 0.52 |
| Maxillary protrusion | 100.0 | 4.1 | 24.7 | 22.2 | 0.3636 | 0.55 |
| Crowding | 48.0 | 75.3 | 68.4 | 40.0 | 0.4364 | 0.62 |

AUC: area under the receiver operating characteristic curve.

**Table 6. Association between deep bite and oral habits.**

| Objective variables | Explanatory variables (questionnaire) | *P*-value | OR (95%CI) |
|---|---|---|---|
| **Deep bite** | Is your child's mouth often open during the day? | 0.8605 | 0.97 (0.65–1.43) |
| | Does your child sleep with their mouth open? | 0.3049 | 0.82 (0.55–1.20) |
| | Does your child have difficulty eating hard foods? | 0.3667 | 0.82 (0.53–1.26) |
| | Does your child often prefer soft foods? | 0.7302 | 0.93 (0.62–1.40) |

95%CI: 95% confidence intervals; OR: odds ratio.

**Table 7. Association between maxillary protrusion and oral habits.**

| Objective variables | Explanatory variables (questionnaire) | *P*-value | OR (95%CI) |
|---|---|---|---|
| **Maxillary protrusion** | Is your child's mouth often open during the day? | **<0.001** | 2.16 (1.45–3.20) |
| | Does your child sleep with their mouth open? | **<0.05** | 1.60 (1.05–2.46) |
| | Does your child have difficulty eating hard foods? | 0.0889 | 0.67 (0.42–1.07) |
| | Does your child often prefer soft foods? | 0.2259 | 0.77 (0.50–1.18) |

95%CI: 95% confidence intervals; OR: odds ratio.

**Table 8. Association between crowding and oral habits.**

| Objective variables | Explanatory variables (questionnaire) | *P*-value | OR (95%CI) |
|---|---|---|---|
| **Crowding** | Is your child's mouth often open during the day? | 0.1440 | 1.37 (0.90–2.08) |
| | Does your child sleep with their mouth open? | 0.2185 | 1.32 (0.85–2.06) |
| | Does your child have difficulty eating hard foods? | **<0.01** | 1.93 (1.24–3.00) |
| | Does your child often prefer soft foods? | **<0.01** | 1.83 (1.20–2.80) |

95%CI: 95% confidence intervals; OR: odds ratio.

## Discussion

This study examined the feasibility of building an AI model that can detect the presence of malocclusion in children in the mixed dentition stage. The results showed that the maxillary protrusion and crowding classification models showed moderate accuracy (AUC > 0.70), and the deep bite classification model showed high accuracy (AUC > 0.90) on the test dataset. In this study, the deep bite classification model showed the highest accuracy. The frontal dental image was the highest contributing factor in the PI of the deep bite classification model. Furthermore, the PI of the questionnaire data was < 1% for all factors, indicating that the questionnaire data were not an important factor in determining deep bite. Furthermore, the chi-square test showed no significant association between the presence of deep bite and the questionnaire data on oral habits (Table 6). No previous study has reported an association between oral habits and deep bite, which is consistent with the findings of this study. These findings indicate that frontal images are a major factor in the construction of a model to determine deep bite. Additionally, the activation map showed that the area of focus in this model was near the bite of the anterior teeth. Since deep bite is determined by the degree to which the maxillary anterior teeth cover the mandibular anterior teeth in clinical practice, it is inferred that a valid algorithm could be created.

In the maxillary protrusion classification model, the frontal image was the most important explanatory variable, followed by the left side image and the right side image. Since the visual determination of maxillary protrusion is performed based on the degree of protrusion of the anterior teeth when viewed from the side, it is considered that the left and right dental

images were also employed as important predictors in this model. Additionally, compared with other models, the contribution of questionnaire data to prediction was higher in this model, and the PI of questionnaire data on the presence or absence of the habit related to incompetent lip seal was 19.0%. In this study, the results of a chi-square test of the association between the presence of maxillary protrusion and questionnaire items related to oral habits showed a significant association between the presence or absence of daytime open mouth and maxillary protrusion (Table 7, $P < 0.001$, OR; 2.16). The incompetent lip seal is a common and frequently observed oral habit in children. A large epidemiological study of pediatric patients aged 3–12 years in Japan showed that approximately 30% of children had the condition [18]. Ohtsugu et al. showed a significant association between the presence or absence of incompetent lip seal and malocclusion in children, and logistic regression analysis to determine malocclusion revealed that the presence or absence of a habit of incompetent lip seal was an important risk factor for malocclusion [17]. Incompetent lip seal with mouth breathing has been reported to lead to abnormal tooth and maxillofacial development, which may affect the health of the dentomaxillofacial system [20]. Children with normal breathing patterns form a sealed oral cavity with the lips closed and the tongue in contact with the palate and lingual side of the upper dentition. Balanced muscle strength between the internal muscles of the tongue and the external muscles of the lips and cheeks is essential for normal development of the maxillary arch. In contrast, keeping the mouth open may result in mouth breathing with the tongue positioned forward or downward, lowering the mandible and potentially causing muscle imbalance. It may result in not only a lack of expansion stimulus to the maxillary side caused by the tongue contact, but also the promotion labial inclination of the maxillary incisors by lack of oral closure, potentially leading to oral cavity and craniofacial anomalies. A cross-sectional study has shown that mouth breathers have significantly more lip closure insufficiency compared to nose breathers, and mouth breathers have significant backward and downward rotation of the mandible and increased maxillary protrusion compared to nose breathers [24]. Furthermore, previous studies have reported a significant association between maxillary protrusion and mouth breathing [25]. These findings indicate that the habit of incompetent lip seal is a risk factor for maxillary protrusion. Based on the above, in the maxillary protrusion classification model, we can assume that a valid algorithm has been constructed, as the factors important for prediction are consistent with those previously reported.

In the crowding classification model, the frontal image was the most important explanatory variable. Since the visual judgment of crowding is based on the degree of overlap between adjacent teeth when viewed from the front, this crowding prediction model was considered valid because the factors that are important in human visual judgment and those that are important in this prediction model are consistent. In addition to images, questionnaire data on mastication habits were also calculated as partly important factors in the classification of crowding (PI: 2.01%) (Fig 6). Crowding is a malocclusion caused by insufficient space for teeth to align properly. It has been suggested that immature jaw development due to the habit of chewing soft foods may contribute to aggravation of crowding. Previous studies have shown that the decline in masticatory ability due to the widespread use of processed foods may cause underdevelopment of the jaw [16]. Insufficient space in the dentition due to underdeveloped jaws has been reported as a contributing factor to the occurrence of crowding [15]. Furthermore, a previous study showed that jaw growth is promoted by the habit of eating hard foods during childhood [14]. In particular, the median palatine suture is a growth site in maxillary expansion and plays a crucial role in the lateral growth of the maxilla. High masticatory loads are suggested to contribute to maxillary growth through stimulation of these sutures. The above animal experiments have reported that in group where loads were reduced by a soft diet, the width of the palatine sutures decreased compared to group fed a hard diet, and returning to a hard diet induced reactivation of the suture region. Based on these findings, it can be inferred that the habit of aversion to eating hard foods is a risk factor for crowding. Furthermore, in this study, the chi-square test of the association between the presence of crowding and the questionnaire item related to the habit of eating hard foods showed a significant association (Table 8, $P < 0.01$, OR = 1.83). These findings indicate that the habit of eating hard foods is an important factor in predicting crowding. Because the crowding classification model used frontal images of teeth and items related to chewing habits which have been reported as risk factors for crowding, it is believed that a valid algorithm has been constructed.

When the performance of the models was evaluated on the new data, the three models scored well on all performance measures, especially in sensitivity. Sensitivity is the most important metric in the early detection of disease. High sensitivity values of 91.2%, 70.0% and 80.0% were obtained for the three models (deep bite, maxillary protrusion and crowding). The sensitivity of the deep bite model was high, whereas that of the maxillary protrusion model was lower than that of the other models. The deep bite model is thought to be easier to learn by machine because the malpositions are concentrated in the anterior teeth, and there are fewer case patterns, whereas the maxillary protrusion model is difficult to learn because the angle of view of the left and right images changes more easily than the frontal images, creating diversity in the images. This may have created a situation where it is difficult to learn. All models showed high performance in recall value, but the precision value were poor for the maxillary protrusion and crowding models. This indicates that although the constructed models have a low false negative rate, they tend to produce a certain number of false positives. Therefore, the current model is not suitable for precise diagnosis, but we think that it is suitable as a screening tool in environments where precise diagnosis is difficult (e.g., health check-ups). In the future, it is suggested that further improvements to the model are necessary to implement it as a screening tool while maintaining a high recall value and improving the precision value.

For comparison, a binary classification model was also created using logistic regression with only the questionnaire data. The results showed that the logistic regression model had poor classification accuracy for all types of malocclusion (AUC < 0.70), suggesting that questionnaire data alone may not be sufficient for accurately classifying malocclusion. Based on these results, it was suggested that machine learning models using dental images are useful for constructing an algorithm for classifying malocclusion. Furthermore, the results of permutation importance results showed that not only dental images but also the presence or absence of these oral habits related to open-mouth posture and chewing habits are important in determining of crowding and maxillary protrusion. However, malocclusion is a multifactorial disease, and other oral habits may also contribute to the development of malocclusion. In this study, we selected specific oral habits that have been suggested to be associated with malocclusion and constructed classification models as exploratory analysis. In the future, we will need to comprehensively collect and analyze questionnaire data while controlling for confounding factors. This will allow us to narrow down the factors that are more strongly associated with specific malocclusions and refine the algorithm. However, It should be noted that information on oral habits to be included in the algorithm should be limited to those that can be identified by parents and general public, such as open mouth posture or chewing habits. Considering the application and convenience of the model, it is necessary to limit the explanatory variables input into the algorithm to items that can be easily determined. Some oral habits, such as tongue thrust swallowing, can only be determined by a specialist dentist. Going forward, we plan to further refine the questionnaire data while considering the limitations of oral habit information that can be input into this algorithm.

AI research on pediatric dentistry is still in its infancy. However, several studies are underway. You et al. reported a CNN model for plaque detection in primary teeth that achieved clinically acceptable performance compared to experienced pediatric dentists [26]. Okazaki et al. constructed an AlexNet model to classify multiple dental abnormalities in children using panoramic radiographs and showed a certain degree of accuracy [27]. Furthremore, to develop the AI tool that supporting dental diagnostics, Erfan et al. designed dental occlusion classification model using intraoral photographs for adult patients [28]. Therefore, these same algorithms could potentially be used to identify mixed dentition in children. However, these studies were conducted using panoramic and intraoral images, such as those taken in a medical setting, and AI was designed to assist dental professionals in efficient diagnosis in dental practices. Furthermore, these models are not checking tools that can make decisions at any location, and whether they can be predicted using images captured at home, such as smartphone images, is unclear. In this study, an algorithm was developed that uses dental images captured by guardians using their smartphones, and the possibility of determining dental alignment with a certain accuracy was confirmed.

This study has several limitations. First, the dentition and occlusal status were determined based on the school dentist's guidelines and previous study [13]. Additionally, this study did not quantitatively evaluate inter-rater and intra-rater

agreement during assessment using metrics such as Cohen's kappa coefficient or intraclass correlation coefficient. In the future study, to enhance the algorithm's accuracy, it is crucial to adopt annotation judgment values provided by experienced orthodontists. Furthermore, it is critically important to build the algorithm based on annotation data demonstrating high inter-rater and/or intra-rater agreement. After constructing an algorithm using annotation data with higher accuracy for training, it is also necessary to compare the prediction results of the developed algorithm with the diagnostic values obtained by multiple orthodontists. This study has not yet verified the degree to which the predicted values from the constructed algorithm align with the diagnostic results of experienced orthodontists. Future research should compare the predictive accuracy of AI with the diagnostic accuracy of experienced orthodontists, build more reliable AI models, and further validate their clinical utility. Second, this study has not sufficiently examined the applicability of data obtained in the real world. Under real-world conditions, the angles and quality of dental images taken may become even more diverse. Therefore, it is necessary to investigate how well performance is maintained when using photographs taken in the real world. This study employed a dataset of dental images captured without restrictions on location or ambient lighting, using only minimal dental guide frames and simple instructions. Furthermore, since dental images were collected from over 500 individuals in their home environments, we believe a certain level of diversity in the collected data was achieved. However, in real-world use, issues with angle diversity may arise even within the defined guide frame, depending on the parent's operational ability and IT literacy. Regarding image quality, while subjects were recruited using smartphones that were mainstream at the time of this study, significantly outdated or minor models were not evaluated. Therefore, in real-world scenarios, image input from a wider variety of devices is assumed. Moving forward, it is necessary to enhance the algorithm's versatility while simultaneously investigating its applicability in real-world scenarios. Moreover, implementing UI logic for selecting input images is crucial. This includes functions to exclude excessively dark images based on color values (e.g., RGB) or images that are significantly out of focus and unclear. Additionally, implementing an instruction interface that is easy for all subjects to understand is required. Third, after constructing a classification model with enhanced accuracy and versatility, validation using external datasets must also be conducted in the future. Leveraging the advantages of the application, we plan to collect images on a larger scale through the application and consider accuracy validation using multiple external datasets beyond the current trial in the future. Finally, it is necessary to confirm whether the early detection of malocclusion using this algorithm can promote changes in patients' health-seeking behavior. In addition, to prevent missed dental visits after self-diagnosis, it is necessary to design a user interface that encourages dental visits after screening and to implement informational screens that promote behavior change. In summary, in addition to improving the accuracy of these algorithms, it is important to consider the design specifications for a screening tool that can effectively promote behavior change in patients in the future.

## Conclusions

We used a smartphone application to collect dental images with aligned angles of view. Based on these images, we developed AI-based binary classification model for malocclusion during the mixed dentition period and verified its potential as a screening tool. The results showed that all models performed well. Further refinement is needed for clinical application, but this study shows that the AI-based binary classification model is a promising tool for screening malocclusion during the mixed dentition stage.

## Author contributions

**Conceptualization:** Kengo Oka.

**Data curation:** Kengo Oka, Saki Uemura.

**Formal analysis:** Kengo Oka, Saki Uemura.

**Funding acquisition:** Kengo Oka, Satoru Morishita, Kei Kurita.

**Investigation:** Kengo Oka, Saki Uemura, Satoru Morishita.

**Methodology:** Kengo Oka, Saki Uemura.

**Project administration:** Kengo Oka, Satoru Morishita.

**Resources:** Kengo Oka, Saki Uemura.

**Supervision:** Satoru Morishita, Kei Kurita, Yukio Yamamoto.

**Writing – original draft:** Kengo Oka.

**Writing – review & editing:** Satoru Morishita, Kei Kurita.

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
