## [Decision Letter · Decision Letter 0]

3 Mar 2025

Dear Dr. Oka,

Thank you for submitting your manuscript to PLOS ONE. After careful consideration, we feel that it has merit but does not fully meet PLOS ONE’s publication criteria as it currently stands. Therefore, we invite you to submit a revised version of the manuscript that addresses the points raised during the review process.

The study question is quite interesting, however there are several aspects, especially methodological ones, to be considered. If you decide to resubmit an adjusted version, please incorporate the comments of the 3 reviewers.

We look forward to receiving your revised manuscript.

Kind regards,

Erika Barbara Abreu Fonseca Thomaz, Ph.D

Academic Editor

PLOS ONE

Journal Requirements:

For additional information about PLOS ONE ethical requirements for human subjects research, please refer to http://journals.plos.org/plosone/s/submission-guidelines#loc-human-subjects-research .

 “This work was funded by Lion Corporation (https://www.lion.co.jp/en/).”

“I have read the journal's policy and the authors of this manuscript have the following competing interests: [The authors of this manuscript employed by LION Corporation. Recruitment of participants was performed by Macromill Inc. (Tokyo, Japan) under consignment from Lion Corporation.]”

We note that one or more of the authors are employed by a commercial company: LION Corporation

6. We note that you have indicated that there are restrictions to data sharing for this study. PLOS only allows data to be available upon request if there are legal or ethical restrictions on sharing data publicly. For more information on unacceptable data access restrictions, please see http://journals.plos.org/plosone/s/data-availability#loc-unacceptable-data-access-restrictions.

Reviewers' comments:

Reviewer's Responses to Questions

**Comments to the Author**

1. Is the manuscript technically sound, and do the data support the conclusions?

Reviewer #1: Partly

Reviewer #2: Partly

Reviewer #3: Partly

2. Has the statistical analysis been performed appropriately and rigorously?

Reviewer #1: Yes

Reviewer #2: No

Reviewer #3: Yes

3. Have the authors made all data underlying the findings in their manuscript fully available?

Reviewer #1: Yes

Reviewer #2: No

Reviewer #3: Yes

4. Is the manuscript presented in an intelligible fashion and written in standard English?

Reviewer #1: Yes

Reviewer #2: Yes

Reviewer #3: No

Reviewer #1: The study addresses a significant public health issue (malocclusion) and explores an innovative AI-based approach for its early detection. Below are some points that need to be improved and clarified in the manuscript.

Abstract

• Clearly specify the type of study in the methods section.

• Indicate where the sample was drawn from.

• When stating that questionnaire data were collected, it is important to describe which specific data were included in the algorithm.

Introduction

The introduction presents an interesting and relevant study topic but requires refinement in terms of epidemiological accuracy, research justification, and methodological clarity.

• While the introduction outlines the prevalence and consequences of malocclusion, it does not explicitly state the core research problem.

• Explain the limitations of current detection methods and introduce AI-based image analysis as a potential solution.

• The statement: “The worldwide prevalence of malocclusion is high (up to 54%)” lacks specificity. Prevalence varies significantly based on age group, ethnicity, and classification criteria (e.g., Angle’s classification vs. other methods). The introduction should: specify the population from which this prevalence estimate originates; differentiate between malocclusion severity levels (mild, moderate, severe); and ensure all epidemiological claims are backed by references from peer-reviewed studies.

• The claim that “patients themselves are unaware that the degree of malocclusion is severe enough to require treatment” lacks supporting evidence. The introduction should provide references or data to support this assertion rather than assuming a singular cause.

• Clearly define the acronym OECD (Organization for Economic Cooperation and Development) when it first appears in the introduction.

• The introduction references AI-based classification but does not outline the type of AI model being considered (e.g., CNNs, deep learning, transfer learning) and the challenges in training AI for malocclusion detection (e.g., dataset bias, image quality issues, need for expert annotations). A brief mention of these challenges would help frame the study’s contribution in a realistic context.

Methods

• The study is described as a pilot study, yet it includes 520 participants. The authors should clarify if this is truly a pilot study or an exploratory AI development study. If it is a pilot study, explain why such a large sample was used, and If it is a full-scale study, provide justification for the sample size based on power calculations.

• The recruitment method via Macromill Inc. (a commercial survey company) may introduce selection bias: Were participants from diverse socioeconomic backgrounds? Were they representative of the general population of Japanese children? How was participant diversity ensured?

• Describe how the sampling method ensures generalizability and address potential selection bias.

• It would be beneficial to add a section detailing quality control measures for smartphone images and any inter-rater agreement measures.

• What specific machine learning (ML) algorithms were used? (e.g., CNNs, decision trees, SVM, etc.)

• How was model selection performed beyond AutoML automation?

• What hyperparameter tuning methods were used?

• The blender models mentioned are ensemble models, but were any baseline models (e.g., logistic regression, traditional orthodontic assessments) used for comparison? How did the chosen models perform against existing AI models in dentistry?

• The study uses AUC-ROC, sensitivity, specificity, and accuracy, but the precision and F1-score are missing, and these are crucial for imbalanced datasets.

• In the statistical analysis, was any adjustment made for potential confounding factors?

Results

• In the results tables, I suggest including p-values instead of n.s. (not significant).

Discussion

• The discussion does not mention whether the model's performance holds up under real-world conditions or external datasets.

• It does not acknowledge how image distortions, lighting conditions, and smartphone differences may have affected model accuracy.

• Address the need for external validation using independent datasets.

• The discussion only focuses on sensitivity and accuracy but does not discuss misclassification cases. Include an error analysis, discussing false positive and false negative rates and explain the potential clinical consequences of these misclassifications.

• The discussion claims that questionnaire data had little impact on overbite predictions but was more relevant for overjet and crowding. However, it does not explain why this variation occurs.

• The relationship between oral habits and malocclusion should be discussed in greater depth, particularly how AI can or cannot capture behavioral influences. Provide a deeper theoretical explanation of why oral habits impact certain malocclusions more than others.

• Address potential dataset biases and the need for diverse, representative training data and discuss ethical concerns regarding AI-based self-diagnosis and whether these models should only be used as a pre-screening tool.

Reviewer #2: This study presents an innovative approach to diagnosing malocclusions in children using artificial intelligence (AI) based on photographs taken by their caregivers. The topic is highly relevant, particularly given the expanding role of AI in Dentistry and the feasibility of remote methods for screening and monitoring.

However, a critical methodological limitation must be addressed: the lack of clinical validation of the AI model by experienced orthodontists. Without this comparison, it is not possible to determine the system’s accuracy or practical applicability. Since orthodontic diagnosis is inherently complex, validation against human experts is essential to ensure that the model correctly identifies malocclusion patterns.

I recommend that the authors consider the following approaches to strengthen the study:

Include a preliminary validation – If feasible, the authors could compare the AI-generated diagnoses with those made by orthodontists in a selected sample.

Additionally, it would be helpful for the authors to clarify:

How the images were standardized to minimize variations (e.g., lighting, capture angle, resolution).

Reviewer #3: I congratulate you on the idea of the article for being original and for the new insights, but some methodological issues regarding the study design and methodology need to be made clearer in the study.

First, it is important to include a subtopic in the methodology that specifies the study design. From the title, “classification algorithm”, it is possible that this study is cross-sectional.

It is important to state in the methodology, perhaps in the description of the analysis, that The structure of the present manuscript followed the guidelines Developing and Reporting Machine Learning Predictive

Models in Biomedical Research.

Many points are not clear in the analysis.The first point is whether variables with missing data were excluded from the analysis and what strategies were adopted to obtain the inputs for developing the algorithm.

The article does not mention whether other classification models were tested, in addition to SVM. If other algorithms such as random forest, KNN, XGboost were tested, it would be essential to mention the methodology and organize a table with the information found for these tested algorithms.

I suggest including, in addition to sensitivity, specificity, AUC, other metrics, such as F-1, accuracy and recall.

It would also be interesting to point out, in addition to the associations, the degree of importance of the input variables. This can be represented by a graph modified by SHapley Additive exPlanations.

Congratulations on the article! I recommend including this information and that the discussion consider these aspects.

**Do you want your identity to be public for this peer review?** For information about this choice, including consent withdrawal, please see our Privacy Policy

Reviewer #1: **Yes: ** Lorena Lucia Costa Ladeira

Reviewer #2: No

Reviewer #3: No

---

## [Author Response · Author response to Decision Letter 1]

16 May 2025

Response to Reviewers

Response to Academic editor:

Thank you for submitting your manuscript to PLOS ONE. After careful consideration, we feel that it has merit but does not fully meet PLOS ONE’s publication criteria as it currently stands. Therefore, we invite you to submit a revised version of the manuscript that addresses the points raised during the review process. The study question is quite interesting, however there are several aspects, especially methodological ones, to be considered. If you decide to resubmit an adjusted version, please incorporate the comments of the 3 reviewers.

RESPONSE:

We sincerely appreciate the valuable and insightful comments of the editors and reviewers. We have revised and improved the manuscript as you have indicated. Below are our responses to the issues raised. We have also highlighted in the “Revised Manuscript with Track Changes” file the areas where we have updated the manuscript in response to the reviewers' comments.

RESPONSE:

Thank you for recommending that we register our protocol on Protocols.io. We hope to publish the results of our study as soon as possible in a peer-reviewed open access journal. However, the protocol registration process is expected to take some time. First, we need to prepare the necessary documents for protocol registration on Protocols.io. After that, we must obtain approval from the Institutional Review Board (IRB) regarding the publication of the protocol. Therefore, we will revisit this matter once the manuscript has been accepted for publication.

Journal Requirements:

RESPONSE:

We have reviewed PLOS ONE's style requirements and have corrected font size, formatting, etc. We made a mistake with the font size in Level 1-3 Heading, so We corrected it to the correct size. We also have changed the format of the formula for the model's evaluation index, as noted on page 20. We hope that these revisions now adhere to the journal’s style requirements.

RESPONSE:

This study utilizes the AI platform provided by Datarobot to create the algorithm, due to restrictions in the license agreement with Datarobot Inc, we are unable to share regarding the code of the algorithm. Therefore, we have added information about the algorithm as much as possible in the Method sections of the text. In the Method section, the subtopic “Modelling and valdation” on page 16 – 17, we have described information on the blue prints of each algorithm we have developed.

P17, lines 3 – P18, lines 2:

The following modeling was performed by Datarobot on autopilot. Questionnaire data and image data were automatically separated from the dataset, and feature extraction from the image data was performed by a pre-trained deep learning model of SqueezeNet, a type of CNN that had been pre-trained. Next, Datarobot used the binary data from questionnaire data and the extracted image features as explanatory variables to create a classification model for the presence or absence of overbite using the Elastic Net Classifier.

In the overjet classification model, before building the model using all data sets, Datatobot only used the feature values extracted by SqueezeNet, and calculated the predicted value for the presence or absence of overjet using the Elastic Net Classifier. Then, Datarobot used the predicted value itself as a new feature value, integrated it with the binary data from the questionnaire data, and constructed an algorithm for judging overjet using the Elastic Net Classifier.

In the crowding classification model, before building the model using all data sets, Datarobot used only the image features extracted by SqueezeNet as explanatory variables and calculated predictions for the presence or absence of crowding using the Elastic Net Classifier. Datatobot adopted the above prediction values themselves as new features. By feeding the binary data from the questionnaire and the prediction values into the SVM Classifier model as explanatory variables, we constructed a crowding classgfcation model.

RESPONSE:

We added more detail on how to provide participant consent. The revisions are as follows.

P10, lines 1–3:

When obtaining consent, a consent form was displayed on the web screen and buttons were provided for subjects to indicate whether they agreed or disagreed.

P10, lines 8–11:

Consent was obtained by placing a response button on the website. Parents or guardians were asked to indicate whether they had obtained consent from their child by pressing the button.

“This work was funded by Lion Corporation (https://www.lion.co.jp/en/).” Please state what role the funders took in the study. If the funders had no role, please state: "The funders had no role in study design, data collection and analysis, decision to publish, or preparation of the manuscript."

If this statement is not correct you must amend it as needed. Please include this amended Role of Funder statement in your cover letter; we will change the online submission form on your behalf.

RESPONSE:

The funding organization (Lion Corporation) did not play any role in the study design, data collection and analysis, decision to publish, or preparation of the manuscript. It only provided financial support in the form of authors’ salaries and/or research materials. Therefore, we stated “The funders had no role in study design, data collection and analysis, decision to publish, or preparation of the manuscript.” in the original Funding Statement. We have also included the above information in the cover letter.

“I have read the journal's policy and the authors of this manuscript have the following competing interests:

[The authors of this manuscript employed by LION Corporation. Recruitment of participants was performed by Macromill Inc. (Tokyo, Japan) under consignment from Lion Corporation.]”

We note that one or more of the authors are employed by a commercial company: LION Corporation

a. Please provide an amended Funding Statement declaring this commercial affiliation, as well as a statement regarding the Role of Funders in your study. If the funding organization did not play a role in the study design, data collection and analysis, decision to publish, or preparation of the manuscript and only provided financial support in the form of authors' salaries and/or research materials, please review your statements relating to the author contributions, and ensure you have specifically and accurately indicated the role(s) that these authors had in your study. You can update author roles in the Author Contributions section of the online submission form. Please also include the following statement within your amended Funding Statement.

“The funder provided support in the form of salaries for authors [insert relevant initials], but did not have any additional role in the study design, data collection and analysis, decision to publish, or preparation of the manuscript. The specific roles of these authors are articulated in the ‘author contributions’ section.” If your commercial affiliation did play a role in your study, please state and explain this role within your updated Funding Statement.

RESPONSE:

As noted above, The funding organization (Lion Corporation) did not play any role in the study design, data collection and analysis, decision to publish, or preparation of the manuscript. Therefore, we have also included the above information in the original Funding Statement and the cover letter. This work was fully funded by Lion Corporation (https://www.lion.co.jp/en/). K Oka, S Uemura, S Morishita, Y Yamamoto, K Kurita are employees of Lion Corporation. Lion Corporation provided support in the form of salaries for authors. We also reviewed the Author Contributions section of the online submission form and confirm that the role(s) that the authors had in this study have been detailed accurately. Therefore, we added to the following sentence to the Funding Statement:

“The funder provided support in the form of salaries for authors [K Oka, S Uemura, S Morishita, Y Yamamoto, K Kurita], but did not have any additional role in the study design, data collection and analysis, decision to publish, or preparation of the manuscript. The specific roles of these authors are articulated in the ‘author contributions’ section.”

Within your Competing Interests Statement, please confirm that this commercial affiliation does not alter your adherence to all PLOS ONE policies on sharing data and materials by including the following statement: "This does not alter our adherence to PLOS ONE policies on sharing data and materials.” (as detailed online in our guide for authors http://journals.plos.org/plosone/s/competing-interests) . If this adherence statement is not accurate and there are restrictions on sharing of data and/or materials, please state these. Please note that we cannot proceed with consideration of your article until this information has been declared. Please include both an updated Funding Statement and Competing Interests Statement in your cover letter. We will change the online submission form on your behalf.

RESPONSE:

There are no changes in the Competing Interests Statement. Declarations related to employment have already been described in the original text. As of today (May 16, 2025), the prediction algorithm built in this study is not marketed as a product of Lion Corporation and there is no licensing status for the algorithm. This information will be added to the Competing Interests Statement as needed. In Competing Interests Statement section, I confirmed that this commercial affiliation does not alter our adherence to all PLOS ONE policies, and we added to the following sentence to the Competing Interests Statement section: "This does not alter our adherence to PLOS ONE policies on sharing data and materials.” We have also included information in our cover letter regarding our most recent Funding Statement and Competing Interests Statement.

6. We note that you have indicated that there are restrictions to data sharing for this study. PLOS only allows data to be available upon request if there are legal or ethical restrictions on sharing data publicly. For more information on unacceptable data access restrictions, please see http://journals.plos.org/plosone/s/data-availability#loc-unacceptable-data-access-restrictions. Before we proceed with your manuscript, please address the following prompts:

b) If there are no restrictions, please upload the minimal anonymized data set necessary to replicate your study findings to a stable, public repository and provide us with the relevant URLs, DOIs, or accession numbers. For a list of recommended repositories, please see https://journals.plos.org/plosone/s/recommended-repositories. You also have the option of uploading the data as Supporting Information files, but we would recommend depositing data directly to a data repository if possible.

RESPONSE:

Data cannot be shared publicly because of a lack of such description in the study protocol. For items not listed in the research protocol, data sharing is restricted by Instituitonal Review Board of LION Corporation. Data are available from the Instituitonal Review Board of LION Corporation for researchers who meet the criteria for access to confidential data. (Contact via Kazuo Mukasa (Instituitonal Review Board member of LION Corporation), E-mail: mukkaz@lion.co.jp.)

RESPONSE: Following the instructions, we have confirmed using PACE that the figures meet the requirements of PLOS One.

Response to Reviewer #1:

The study addresses a significant public health issue (malocclusion) and explores an innovative AI-based approach for its early detection. Below are some points that need to be improved and clarified in the manuscript.

RESPONSE:

We would

---

## [Decision Letter · Decision Letter 1]

30 Jul 2025

Dear Dr. Oka,

Thank you for submitting your manuscript to PLOS ONE. After careful consideration, we feel that it has merit but does not fully meet PLOS ONE’s publication criteria as it currently stands. Therefore, we invite you to submit a revised version of the manuscript that addresses the points raised during the review process.

Please thoroughly address the comments provided by Reviewers 2 & 4.

We look forward to receiving your revised manuscript.

Kind regards,

Boyen Huang, DDS, MHA, PhD

Academic Editor

PLOS ONE

Journal Requirements:

Reviewers' comments:

Reviewer's Responses to Questions

**Comments to the Author**

Reviewer #1: All comments have been addressed

Reviewer #2: (No Response)

Reviewer #4: (No Response)

2. Is the manuscript technically sound, and do the data support the conclusions?

Reviewer #1: Yes

Reviewer #2: Yes

Reviewer #4: Partly

3. Has the statistical analysis been performed appropriately and rigorously?

Reviewer #1: Yes

Reviewer #2: Yes

Reviewer #4: No

4. Have the authors made all data underlying the findings in their manuscript fully available?

Reviewer #1: Yes

Reviewer #2: Yes

Reviewer #4: No

5. Is the manuscript presented in an intelligible fashion and written in standard English?

Reviewer #1: Yes

Reviewer #2: Yes

Reviewer #4: Yes

Reviewer #1: Thank you for the opportunity to review the revised manuscript titled "Accuracy of AI-based binary classification for detecting malocclusion in the mixed dentition stage." The authors have thoroughly addressed the previous comments and substantially improved the manuscript in both content and clarity.

The rationale and relevance of the study are now well established, with improved contextualization of the epidemiology of malocclusion and the limitations of current screening tools. The methodological section has been clarified, particularly regarding the AI modeling process, dataset composition, and participant recruitment strategy. Importantly, the addition of precision and F1 scores strengthens the evaluation of the models given the potential for class imbalance.

The discussion now appropriately acknowledges the model's limitations, including the lack of external validation, potential device variability, and the non-diagnostic nature of the tool. The inclusion of logistic regression as a baseline was also a valuable addition.

Overall, the manuscript now meets the standards for publication. I recommend acceptance with minor revisions, limited to emphasizing in the conclusion that the AI model is intended as a screening tool and not a substitute for clinical diagnosis.

Reviewer #2: I thank the authors for the clarifications provided in the revised manuscript. The use of smartphone images for orthodontic screening is indeed promising, and the study addresses a relevant and timely topic.

That said, some important aspects still require clarification:

Introduction

The Introduction remains overly long, particularly with the additions made in the latest revision. I recommend streamlining this section to focus more directly on the study rationale and objectives.

Data annotation process

The authors mention that three researchers jointly reviewed sample images of malocclusion and calibrated their judgments prior to the annotation process. While this is a positive step, the description remains vague and insufficiently detailed. To enhance transparency and reproducibility, I recommend that the authors clarify:

How the calibration was conducted (e.g., through consensus rounds, training sessions, or use of reference cases).

The specific diagnostic criteria used to define each type of malocclusion .

Whether inter- and/or intra-rater agreement was quantitatively assessed (e.g., Cohen’s kappa, intraclass correlation coefficients). If not, this limitation should be explicitly acknowledged in the Discussion section.

Results – Table Footnote

Please consider removing the abbreviation “n.s.: not significant” from the table footnote, as this abbreviation does not appear to be used in the table itself.

Discussion section

The two recently added paragraphs (“Third” and “Fourth”) regarding algorithm accuracy are somewhat repetitive. I suggest consolidating them to improve clarity and conciseness in the Discussion.

Reviewer #4: The use of AI in detecting malocclusion is an interesting topic that warrants investigation. However, in my opinion, several elements of this work need to be improved or clarified.

Major concern:

1. Data Annotation, lines 222-33. It is mentioned that the images were annotated 'simultaneously' by three researchers. However, it is not clear if they annotated independently or together. Did they discuss? How did they reach the final conclusion when disagreement existed?

2. Modeling and validation, lines 266-86. Although some details of the algorithms used were provided after the first revision, it is unclear why different methods/ models were applied to different conditions (overbite, overjet, crowding). Clarifications are needed.

3. Related to the above concern, since the predicted values using the image features from Elastic Net Classifiers were used as explanatory variables for overjet and crowding, would this have impacted the importance of the images? A comparison between this model and a model using the image features (instead of predicted values from the image features) may ease this concern.

4. The values of conducting both chi-sq test (lines 340 - 48) and the logistic regression (lines 328-337) are unclear. In fact, chi-sq test can only check the association between the response variable and one single explanatory variable (Q1 - Q4) at a time, while logistic regression can assess the relationship between the response and all explanatory variables. From the logistic regression model, adjusted odds ratios can be obtained as well. In my opinion, there is no need to do both.

5. Lines 410-33. In presenting the results, there is no need to mention every metric.

6. Lines 438-47. The associations between the presence of conditions and the questionnaire questions are important. More details should be provided. For example, instead of 'there is a significant association', more details can be provided - whether some habits increase/ reduce the risk of developing a certain condition?

Minor comments:

1. Lines 34 - meaning of 'Q1-Q4' is not clear in the abstract. Suggest replacing with some simple descriptions.

2. Lines 163-4 - clarify if the participant was 'dropped out' or 'excluded'.

3. Lines 240-44. These sentences seem to repeat the previous ones.

4. Lines 244-47. Rewrite this sentence.

5. Line 256. Either "...used in the algorithm" or "...used in the analysis", but not "algorithm analysis"

6. Line 266. "Using the training dataset", not "80% of the training dataset". Similar in Line 313.

7. Line 286. "classification"

8. Line 292. "select" instead of "selected"

**Do you want your identity to be public for this peer review?** For information about this choice, including consent withdrawal, please see our Privacy Policy

Reviewer #1: **Yes: ** Lorena Lucia Costa Ladeira

Reviewer #2: No

Reviewer #4: No

---

## [Author Response · Author response to Decision Letter 2]

9 Sep 2025

Response to Academic editor:

Thank you for submitting your manuscript to PLOS ONE. After careful consideration, we feel that it has merit but does not fully meet PLOS ONE’s publication criteria as it currently stands. Therefore, we invite you to submit a revised version of the manuscript that addresses the points raised during the review process. Please thoroughly address the comments provided by Reviewers 2 & 4. Please submit your revised manuscript by Sep 13 2025 11:59PM. If you will need more time than this to complete your revisions, please reply to this message or contact the journal office at plosone@plos.org. Please include the following items when submitting your revised manuscript:

・A rebuttal letter that responds to each point raised by the academic editor and reviewer(s). You should upload this letter as a separate file labeled 'Response to Reviewers'.

・A marked-up copy of your manuscript that highlights changes made to the original version. You should upload this as a separate file labeled 'Revised Manuscript with Track Changes'.

・An unmarked version of your revised paper without tracked changes. You should upload this as a separate file labeled 'Manuscript'.

RESPONSE:

We would like to express our sincere gratitude to the editors and reviewers for their valuable comments and suggestions. We have revised and improved the manuscript as you have indicated. Below are our responses to the issues raised. We have also highlighted in the “Revised Manuscript with Track Changes” file the areas where we have updated the manuscript in response to the reviewers' comments. Please note that there are no changes to the financial disclosure from the previous submission.

If applicable, we recommend that you deposit your laboratory protocols in protocols.io to enhance the reproducibility of your results. Protocols.io assigns your protocol its own identifier (DOI) so that it can be cited independently in the future. For instructions see: https://journals.plos.org/plosone/s/submission-guidelines#loc-laboratory-protocols. Additionally, PLOS ONE offers an option for publishing peer-reviewed Lab Protocol articles, which describe protocols hosted on protocols.io. Read more information on sharing protocols at https://plos.org/protocols?utm_medium=editorial-email&utm_source=authorletters&utm_campaign=protocols.　

We look forward to receiving your revised manuscript.

RESPONSE:

Thank you for recommending that we register our protocol on Protocols.io. We hope to publish the results of our study as soon as possible in a peer-reviewed open access journal. However, the protocol registration process is expected to take some time. First, we need to prepare the necessary documents for protocol registration on Protocols.io. After that, we must obtain approval from the Institutional Review Board (IRB) regarding the publication of the protocol. Therefore, we will revisit this matter once the manuscript has been accepted for publication.

Response to Reviewer #1:

Thank you for the opportunity to review the revised manuscript titled "Accuracy of AI-based binary classification for detecting malocclusion in the mixed dentition stage." The authors have thoroughly addressed the previous comments and substantially improved the manuscript in both content and clarity.

The rationale and relevance of the study are now well established, with improved contextualization of the epidemiology of malocclusion and the limitations of current screening tools. The methodological section has been clarified, particularly regarding the AI modeling process, dataset composition, and participant recruitment strategy. Importantly, the addition of precision and F1 scores strengthens the evaluation of the models given the potential for class imbalance.

The discussion now appropriately acknowledges the model's limitations, including the lack of external validation, potential device variability, and the non-diagnostic nature of the tool. The inclusion of logistic regression as a baseline was also a valuable addition.

Overall, the manuscript now meets the standards for publication. I recommend acceptance with minor revisions, limited to emphasizing in the conclusion that the AI model is intended as a screening tool and not a substitute for clinical diagnosis.

RESPONSE:

We would like to express our gratitude to you for your insightful comments and suggestions, which have helped us improve our manuscript considerably. In conclusion section, we emphasized that the AI model is intended as a screening tool and not a substitute for clinical diagnosis. Our responses to the comments are provided below.

P3, lines 53 – lines 54

For the detection of malocclusion in mixed dentition, AI-based binary classification models are a promising approach as a screening tool.

P43, lines 747 – P44, lines 752

We used a smartphone application to collect dental images with aligned angles of view and used these images to build a binary classification model of malocclusion in the mixed dentition period using AI.

We used a smartphone application to collect dental images with aligned angles of view. Based on these images, we developed AI-based binary classification model for malocclusion during the mixed dentition period and verified its potential as a screening tool.

P44, lines 753 – P44, lines 757

Although further refinement is needed for clinical application, this study shows that the AI-based binary classification model is a promising approach to detect malocclusion in the mixed dentition period.

Further refinement is needed for clinical application, but this study shows that the AI-based binary classification model is a promising tool for screening malocclusion during the mixed dentition stage.

Response to Reviewer #2:

I thank the authors for the clarifications provided in the revised manuscript. The use of smartphone images for orthodontic screening is indeed promising, and the study addresses a relevant and timely topic. That said, some important aspects still require clarification:

RESPONSE:

We sincerely appreciate your insightful comments and suggestions, which have helped us to significantly improve our manuscript. Our responses to the comments are provided below.

Introduction

The Introduction remains overly long, particularly with the additions made in the latest revision. I recommend streamlining this section to focus more directly on the study rationale and objectives.

RESPONSE:

As you noted, the introduction was too long, so we updated it to emphasize the key points. Also, we deleted the redundant text. The revisions are as follows.

P4, lines 57 – lines 62

The prevalence of malocclusion is high worldwide, and its incidence rate remains at a high level from the deciduous dentition stage to the permanent dentition stage. Alhammadi et al. conducted a systematic review of 53 studies published by November 2016 that met the evaluation criteria and calculated the prevalence of malocclusion [1].

A systematic review by Alhammadi et al. revealed a high global prevalence of malocclusion [1].

P5, lines 73 – P6, lines 97

Castellote et al. found that the perception of malocclusion varies between individuals and between patients and practitioners [4]. Using frontal intraoral photographs of 24 cases classified by the DAI index, they examined the perception of dental aesthetics among dentists, orthodontists, and the general population. The study showed that there were significant differences in IOTN-AC scores between orthodontists, general dentists, and the general population, with orthodontists giving the most severe scores. This study also suggests that there may be a discrepancy between orthodontists and patients in their perception of the need for orthodontic treatment. In addition, Fujimura et al. compared the students' own perception of dental aesthetics and the results of the dentists' assessment using the IOTN-AC scale in 173 Japanese university students [5]. A comparison between the two showed a large discrepancy. Of the 33 students judged to need orthodontic treatment by the dentists, only 3 (9.1%) were judged to need orthodontic treatment by the students' self-assessment. These studies suggest that some patients themselves may not be aware that the degree of malocclusion is severe enough to require treatment.

Castellote et al. found that the perception of malocclusion varies among individuals, patients, and practitioners [4]. They examined the perception of dental aesthetics among orthodontists, general dentists, and the general population by using frontal intraoral photographs cases classified by the Dental Aesthetic Index (DAI). The results revealed significant differences in Index of Orthodontic Treatment Need-Aesthetic Component (IOTN-AC) scores, with orthodontists giving the most severe scores. Another study compared the self-perception of dental aesthetics among 173 Japanese university students with dentists' assessments using the IOTN-AC scale [5]. The comparison showed a large discrepancy: Of the 33 students judged by dentists to need treatment, only 3 (9.1%) believed they need it. These studies suggest that some patients may not be aware that their malocclusion is severe enough to require treatment.

P6, lines 106 – P7, lines 117

While patients visiting a dentist's office are diagnosed with malocclusion through a detailed examination such as head x-rays, in the field of health checkups and epidemiological studies, such as Dental Aesthetic Index (DAI) and Index of Orthodontic Treatment Need (IOTN) are used to detect malocclusion because precise examinations cannot be performed in the site. These are useful screening tools that quantify the need for orthodontic treatment by incorporating function, morphology, and dentition aesthetics into the evaluation criteria, but they are composed of many items and require a large amount of time for evaluation.

Malocclusion is diagnosed through detailed examinations, such as X-rays. However, in large-scale health checkups and epidemiological research, methods such as DAI and IOTN are used to screen for malocclusion. While these indices are useful for quantifying the need for orthodontic treatment, they are time-consuming and may result in inter-rater variability depending on the evaluator.

P7, lines 117 – lines 119

Also, it has been reported that the perception of malocclusion varies among evaluators and patients themselves [4].

P8, lines 127 – lines 138

Based on these reports, it is anticipated that algorithms can be developed to enable simple evaluation of oral health by maximizing the use of oral images and other machine learning methods such as CNN, thereby allowing patients to receive appropriate feedback regardless of location. Furthermore, by utilizing these algorithms as screening tools, it is expected that patients' awareness of visiting clinics will increase, enabling screening for oral diseases even in places where detailed examinations are difficult, such as health checkup venues or home.

Based on these reports, it is anticipated that we can develop algorithms for a simple oral health evaluation by using oral images and machine learning (e.g., CNN). This would not only provide patients with appropriate feedback regardless of their location but also serve as a screening tool to promote clinics visits and enable the detection of oral diseases in places where detailed examinations are difficult, such as at home or during health checkups.

Data annotation process

The authors mention that three researchers jointly reviewed sample images of malocclusion and calibrated their judgments prior to the annotation process. While this is a positive step, the description remains vague and insufficiently detailed. To enhance transparency and reproducibility, I recommend that the authors clarify:

・How the calibration was conducted (e.g., through consensus rounds, training sessions, or use of reference cases).

・The specific diagnostic criteria used to define each type of malocclusion .

RESPONSE:

Thank you for your insightful comments. We have provided detailed information on the data annotation process, including how calibration was performed. Furthermore, there was written mistake in the numerical values in the breakdown, so we have corrected them to the correct values. The corrections are as follows:

P15, lines 254 – 258

The criteria for annotation of each type of malocclusion are as follows:

Overbite: more than half of clinical crown in the lower front teeth are covered by the upper front teeth, Crowding: adjacent teeth overlap by more than 1/4 of the width of the crown, Overjet: overjet is large (approximately 4 mm or more), indicating protrusion of the upper front teeth.

P15, lines 259 – P16, lines 283

After that, three researchers simultaneously evaluated whether or not each subject had malocclusion. Before annotation, sample images of malocclusion were reviewed, and the three researchers calibrated their judgments before beginning annotation. Binary data on the presence and absence of dental malocclusion (overbite, overjet, and crowding) were obtained from smartphone dental images. Those with no clearly imaged dentition were excluded from the analysis before annotation. The breakdown of each malocclusion data after data annotation is shown below. overbite (n=166) / no overbite (n=342), overjet (n=147) / no overjet (n=368), crowding (n=166) / no crowding (n=342).

Based on the above criteria, three researchers annotated the data according to the following procedure. (1) The three researchers simultaneously reviewed reference case photos corresponding to each type of malocclusion (overbite, overjet, and crowding) and calibrated their judgments. (2) Based on each dental image dataset, each researcher independently annotated whether each subject had the corresponding type of malocclusion. (3) The three researchers simultaneously reviewed the annotation data assigned to each of them, and they adopted the value for cases where all three researchers made the same judgment. (4) When the three researchers had different opinions, they reviewed the reference case photos again and re-calibrated their judgments. (5) For subjects with differing judgments, each researcher each re-annotated the data. (6) The three researchers simultaneously reviewed the re-evaluated annotation data. Identical judgments were adopted, and cases with remaining discrepancies were deemed unevaluable and excluded from the dataset. The final breakdown of the dataset regarding the presence of malocclusion after data annotation is as follows: Dataset for overbite (n=508) : overbite (n=166) / no overbite (n=342), Dataset for overjet (n=515) : overjet (n=147) / no overjet (n=368), Dataset for crowding (n=518) : crowding (n=123) / no crowding (n=395).

Whether inter- and/or intra-rater agreement was quantitatively assessed (e.g., Cohen’s kappa, intraclass correlation coefficients). If not, this limitation should be explicitly acknowledged in the Discussion section.

RESPONSE:

Thank you for your valuable comments. This study did not quantitatively assess inter- or intra-rater agreement for data annotation. Therefore, we have mentioned this as a limitation discussed in the Discussion section. The revisions are detailed in the discussion section on the next page.

Results – Table Footnote

Please consider removing the abbreviation “n.s.: not significant” from the table footnote, as this abbreviation does not appear to be used in the table itself.

RESPONSE:

Thank you very much for pointing this out. We have removed this abbreviation from the footnotes in the table. We apologize for this written mistake.

Discussion section

The two recently added paragraphs (“Third” and “Fourth”) regarding algorithm accuracy are somewhat repetitive. I suggest consolidating them to improve clarity and conciseness in the Discussi

---

## [Decision Letter · Decision Letter 2]

30 Sep 2025

Dear Dr. Oka,

Thank you for submitting your manuscript to PLOS ONE. After careful consideration, we feel that it has merit but does not fully meet PLOS ONE’s publication criteria as it currently stands. Therefore, we invite you to submit a revised version of the manuscript that addresses the points raised during the review process.

In addition to fixing the typos and presenting format, please address Reviewer 5's comments on the diagnostic criteria, inclusion/exclusion criteria, and limitations of detection accuracy.

We look forward to receiving your revised manuscript.

Kind regards,

Boyen Huang, DDS, MHA, PhD

Academic Editor

PLOS ONE

Journal Requirements:

Reviewers' comments:

Reviewer's Responses to Questions

**Comments to the Author**

Reviewer #4: All comments have been addressed

Reviewer #5: (No Response)

2. Is the manuscript technically sound, and do the data support the conclusions?

Reviewer #4: Yes

Reviewer #5: Partly

3. Has the statistical analysis been performed appropriately and rigorously?

Reviewer #4: Yes

Reviewer #5: Yes

4. Have the authors made all data underlying the findings in their manuscript fully available?

Reviewer #4: No

Reviewer #5: Yes

5. Is the manuscript presented in an intelligible fashion and written in standard English?

Reviewer #4: Yes

Reviewer #5: No

Reviewer #4: Thank you for putting the effort in revising this manuscript. I believe the paper can be accepted after a careful proofread. Some suggestions/ grammatical errors I spotted include:

Lines 256-258: 'Dataset' should not be capitalized after a comma.

Line 274: suggest removing 'for new data'. Same for line 349.

In the results section, suggest not including the whole questionnaire questions to improve readability (e.g., in lines 451-452, simply 'Questions 1 and 2' would be sufficient.)

Reviewer #5: This is an interesting study that explores the use of artificial intelligence (AI) to detect malocclusion in children during the mixed dentition phase, based on three intraoral photographs (frontal, left, and right views) taken by parents. While the concept is promising and relevant, there are several important issues that need to be addressed:

1. Figure Legends:

The figure legends are embedded directly within the main text. To my knowledge, most academic journals typically require figure legends to be placed in a separate section at the end of the manuscript, not within the main body. I recommend the authors double-check the journal’s formatting guidelines and revise accordingly.

2. Tables:

Similarly, the tables are also embedded within the main text. Again, it is unclear whether PLOS ONE allows this format. Typically, tables should be presented separately, often on individual pages at the end of the manuscript. Please confirm with the journal’s formatting requirements and revise if necessary.

3. Diagnostic Criteria:

On page 13, the authors state:

“The criteria for annotation of each type of malocclusion are as follows: Overbite: more than half of clinical crown in the lower front teeth are covered by the upper front teeth, Crowding: adjacent teeth overlap by more than 1/4 of the width of the crown, Overjet: overjet is large (approximately 4 mm or more), indicating protrusion of the upper front teeth.”

These criteria seem to lack comprehensiveness. What about cases of open bite or anterior crossbite? These types of malocclusion also require treatment and may be even more clinically significant than deep overbite or excessive overjet. The current classification system used by the authors may exclude these important cases, which represents a major methodological limitation.

4. Terminology – “No Overjet” / “No Overbite”:

On page 14, the authors write:

“Dataset for overbite (n=508): overbite (n=166) / no overbite (n=342), Dataset for overjet (n=515): overjet (n=147) / no overjet (n=368), Dataset for crowding (n=518): crowding (n=123) / no crowding (n=395).”

The repeated use of expressions such as “no overjet” and “no overbite” is problematic. Overjet and overbite are quantitative features; it is not accurate to say a person has “no overjet” since everyone has some degree of overjet, whether it is positive or negative. It would be more appropriate to use terms such as “normal overjet” vs. “abnormal overjet,” and similarly for overbite. The manuscript contains many such instances that need to be revised.

Moreover, even if the terminology is corrected, it still does not address the concern raised in point 3. What about cases with insufficient overbite or overjet? Based on the current dataset, it seems the authors have only included cases with excessive overjet or deep bite. This makes the scope of the AI detection system quite narrow. How would the model perform with open bite or anterior crossbite cases? Would the algorithm still be effective?

5. Presentation of Demographic Data:

On pages 21–22, the section describing “Residential area” and “Annual household income” is written in incomplete sentences and lacks clarity. Including this in the main body of the text in its current form is awkward. It may be better to present this data using a figure or a well-formatted table to improve readability.

6. Inclusion/Exclusion Criteria – Figure 3(c):

In Figure 3(c), the child appears to be biting towards the right side with a slight forward posture. According to the authors’ own inclusion/exclusion criteria, this case should have been excluded. This raises concerns about the consistency in applying the inclusion criteria.

7. Limitations of Detection Accuracy:

The manuscript reports that:

“The overjet and crowding classification models showed moderate accuracy (AUC > 0.70).”

This result is somewhat expected. Since the photographs provided by parents are limited to frontal, left, and right views, there are inherent limitations in identifying certain conditions, especially crowding. For example, in deep bite cases, the lower anterior teeth may be obscured in frontal and lateral photos, making it difficult for AI to detect crowding in the lower arch. A photo taken with occlusal view would be far more informative for evaluating crowding, but it is not provided in this study.

**Do you want your identity to be public for this peer review?** For information about this choice, including consent withdrawal, please see our Privacy Policy

Reviewer #4: No

Reviewer #5: No

---

## [Author Response · Author response to Decision Letter 3]

31 Oct 2025

Response to Academic editor:

Thank you for submitting your manuscript to PLOS ONE. After careful consideration, we feel that it has merit but does not fully meet PLOS ONE’s publication criteria as it currently stands. Therefore, we invite you to submit a revised version of the manuscript that addresses the points raised during the review process. In addition to fixing the typos and presenting format, please address Reviewer 5's comments on the diagnostic criteria, inclusion/exclusion criteria, and limitations of detection accuracy. Please submit your revised manuscript by Nov 14 2025 11:59PM. If you will need more time than this to complete your revisions, please reply to this message or contact the journal office at plosone@plos.org. Please include the following items when submitting your revised manuscript:

RESPONSE:

We would like to express our sincere gratitude to the editors and reviewers for their valuable comments and suggestions. We have revised and improved the manuscript as you have indicated. Below are our responses to the issues raised. We have also highlighted in the “Revised Manuscript with Track Changes” file the areas where we have updated the manuscript in response to the reviewers' comments. Please note that there are no changes to the financial disclosure from the previous submission.

If applicable, we recommend that you deposit your laboratory protocols in protocols.io to enhance the reproducibility of your results. Protocols.io assigns your protocol its own identifier (DOI) so that it can be cited independently in the future. For instructions see: https://journals.plos.org/plosone/s/submission-guidelines#loc-laboratory-protocols. Additionally, PLOS ONE offers an option for publishing peer-reviewed Lab Protocol articles, which describe protocols hosted on protocols.io. Read more information on sharing protocols at https://plos.org/protocols?utm_medium=editorial-email&utm_source=authorletters&utm_campaign=protocols.　We look forward to receiving your revised manuscript.

RESPONSE:

Thank you for recommending that we register our protocol on Protocols.io. We hope to publish the results of our study as soon as possible in a peer-reviewed open access journal. However, the protocol registration process is expected to take some time. First, we need to prepare the necessary documents for protocol registration on Protocols.io. After that, we must obtain approval from the Institutional Review Board (IRB) regarding the publication of the protocol. Therefore, we will revisit this matter once the manuscript has been accepted for publication.

Response to Reviewer #4:

Thank you for putting the effort in revising this manuscript. I believe the paper can be accepted after a careful proofread. Some suggestions/ grammatical errors I spotted include:

・Lines 256-258: 'Dataset' should not be capitalized after a comma.

・Line 274: suggest removing 'for new data'. Same for line 349.

・In the results section, suggest not including the whole questionnaire questions to improve readability (e.g., 　　

in lines 451-452, simply 'Questions 1 and 2' would be sufficient.

RESPONSE:

We would like to express our gratitude to you for your insightful comments and suggestions, which have helped us improve our manuscript considerably. We have made the following corrections to the suggestions and grammatical errors you pointed out.

P14, lines 246 – lines 249

Dataset for deep bite (n=508): deep bite (n=166) / no deep bite (n=342), Ddataset for maxillary protrusion (n=515): maxillary protrusion (n=147) / no maxillary protrusion (n=368), Ddataset for crowding (n=518): crowding (n=123) / no crowding (n=395).

P15, lines 264 – lines 267

Dataset for deep bite (n=463): deep bite (n=155) / no deep bite (n=308), Ddataset for maxillary protrusion (n=467): maxillary protrusion (n=138) / no maxillary protrusion (n=329), Ddataset for crowding (n=493): crowding (n=112) / no crowding (n=381).

P16, lines 282 – lines 284

Of the total data, 80% was randomly split as a training dataset for algorithm construction and the remaining 20% as a validation dataset for new data.

P21, lines 358 – lines 360

We used 80% of the data as a training dataset for model construction and the remaining 20% as a validation dataset for new data.

P24, lines 411 – lines 416

The frontal image (PI: 100%) was the most important explanatory variable in the model, followed by the left side image (PI: 91.4%), right side image (PI: 64.7%), questionnaire data on habits related to incompetent lip seal (Question 1Q1: Is your child’s mouth often open during the day?) (PI: 19.0%), and questionnaire data on the habit of eating hard foods (Question 3Q3: Does your child have difficulty eating hard foods) (PI: 4.21%) (Fig 5a).

P25, lines 427 – lines 430

The frontal image (PI: 100%) was the most important explanatory variable in the model, followed by the right side image (PI: 2.32%) and questionnaire data on the habit of eating hard foods (Question 4Q4: Does your child often prefer soft foods?) (PI: 2.01%) (Fig 6a)

P28, lines 471 – lines 473

However, oral habits related to incompetent lip seal (Questions 1 and 2: Is your child’s mouth often open during the day? Question 2: Does your child sleep with their mouth open?) recorded significantly ~

P28, lines 477 – PY, lines 479

Furthermore, oral habits related to avoiding hard foods (Questions 3 and 4: Does your child have difficulty eating hard foods? Question 4: Does your child often prefer soft foods?) recorded significantly higher frequency ~

Response to Reviewer #5:

This is an interesting study that explores the use of artificial intelligence (AI) to detect malocclusion in children during the mixed dentition phase, based on three intraoral photographs (frontal, left, and right views) taken by parents. While the concept is promising and relevant, there are several important issues that need to be addressed:

RESPONSE:

We sincerely appreciate your insightful comments and suggestions, which have helped us to significantly improve our manuscript. Our responses to the comments are provided below.

1. Figure Legends: The figure legends are embedded directly within the main text. To my knowledge, most academic journals typically require figure legends to be placed in a separate section at the end of the manuscript, not within the main body. I recommend the authors double-check the journal’s formatting guidelines and revise accordingly.

2. Tables:

Similarly, the tables are also embedded within the main text. Again, it is unclear whether PLOS ONE allows this format. Typically, tables should be presented separately, often on individual pages at the end of the manuscript. Please confirm with the journal’s formatting requirements and revise if necessary.

RESPONSE:

Thank you for your comments. Regarding figure and table descriptions, we have placed them in the paragraph immediately following the citation within the main text, in accordance with the PLOS ONE guidelines. Therefore, we will maintain the current format. We have also reviewed the PLOS ONE formatting guidelines again to confirm they are correct.　

3. Diagnostic Criteria: On page 13, the authors state: “The criteria for annotation of each type of malocclusion are as follows: Overbite: more than half of clinical crown in the lower front teeth are covered by the upper front teeth, Crowding: adjacent teeth overlap by more than 1/4 of the width of the crown, Overjet: overjet is large (approximately 4 mm or more), indicating protrusion of the upper front teeth.” These criteria seem to lack comprehensiveness. What about cases of open bite or anterior crossbite? These types of malocclusion also require treatment and may be even more clinically significant than deep overbite or excessive overjet. The current classification system used by the authors may exclude these important cases, which represents a major methodological limitation.

RESPONSE:

Thank you for your insightful comments. This study similarly evaluated the presence of other malocclusions important from a dental perspective—such as open bite, crossbite, mandibular protrusion etc. —in addition to maxillary protrusion, crowding, and deep bite. The presence of these malocclusions was determined based on the School Dentist Guidelines and prior studies referenced in the text. In this study, we comprehensively annotated the presence of each malocclusion. Among these, we focused on deep overbite, maxillary protrusion, and crowding to construct three binary classification models.

For example, a case annotated as “0” for maxillary protrusion indicates the absence of maxillary protrusion according to the above criteria. This does not necessarily imply the absence of other malocclusions (a case annotated as “0” for maxillary protrusion may still be annotated as “1” for crossbite, for example). Furthermore, even cases annotated as “1” for maxillary protrusion do not imply the absence of other malocclusions and may include coexisting conditions such as deep bite. Therefore, both the “0” and “1” datasets used in the binary classification model constructed here contain diverse dental data. Within this diverse dataset, we developed algorithm to determine the presence or absence of specific types of malocclusion. To clarify, we have made the following corrections.

P13, lines 223 – lines 231

The dentition and occlusal status of children were comprehensively determined for the presence of each type of malocclusion

from smartphone dental images based on the School Dentist Guidelines of the Japan Association of School Dentists and previous study [13]. The criteria for annotation of each type of malocclusion The annotation criteria for each malocclusion type (deep bite, maxillary protrusion, crowding) used in constructing the three algorithms in this study are as follows: Deep bite: more than half of clinical crown in the lower front teeth are covered by the upper front teeth, crowding: adjacent teeth overlap by more than 1/4 of the width of the crown, maxillary protrusion: overjet is large (approximately 4 mm or more), indicating protrusion of the upper front teeth.

As you pointed out, dental treatment is extremely important for conditions such as open bite, and we believe the development of screening tools for these conditions is also necessary. Moving forward, we plan to conduct research aimed at constructing algorithms capable of binary classification for conditions such as open bite, crossbite, mandibular protrusion, and other malocclusions. In Japan, the prevalence of these malocclusions tends to be lower compared to dental crowding or maxillary protrusion. Among the 519 subjects recruited for this study, only a portion exhibited these specific dental irregularities. When constructing algorithms and performing machine learning, it is necessary to secure a sufficient number of cases with specific malocclusion types. For malocclusions like open bite and crossbite, we need to collect more cases to create the algorithm. This time, as feasibility study, we focused on major malocclusions with higher prevalence rates and explored algorithm construction. We plan to develop a classification algorithm capable of handling all types of malocclusion in the future. Furthermore, we intend to build an ensemble model where algorithms for binary classification of each malocclusion type are connected in parallel. This ensemble model would enable simultaneous risk assessment for multiple malocclusion types with a single input of images and questionnaires.

4. Terminology – “No Overjet” / “No Overbite”:

On page 14, the authors write: “Dataset for overbite (n=508): overbite (n=166) / no overbite (n=342), Dataset for overjet (n=515): overjet (n=147) / no overjet (n=368), Dataset for crowding (n=518): crowding (n=123) / no crowding (n=395).”

The repeated use of expressions such as “no overjet” and “no overbite” is problematic. Overjet and overbite are quantitative features; it is not accurate to say a person has “no overjet” since everyone has some degree of overjet, whether it is positive or negative. It would be more appropriate to use terms such as “normal overjet” vs. “abnormal overjet,” and similarly for overbite. The manuscript contains many such instances that need to be revised.

RESPONSE:

We sincerely apologize for any confusion caused by the incorrect expression. Following the criteria outlined in the main text, we have classified these into two categories: large overjet or no large overjet, as well as large overbite or no large overbite. As you pointed out, to avoid confusion, we have revised the terminology in the main text as follows: “overjet” has been changed to “maxillary protrusion,” and ‘overbite’ has been changed to “deep bite.” Furthermore, in line with the terminology changes, we have revised not only the main text but also the terminology used in Figure 1.

Moreover, even if the terminology is corrected, it still does not address the concern raised in point 3. What about cases with insufficient overbite or overjet? Based on the current dataset, it seems the authors have only included cases with excessive overjet or deep bite. This makes the scope of the AI detection system quite narrow. How would the model perform with open bite or anterior crossbite cases? Would the algorithm still be effective?

RESPONSE:

Thank you for your insightful comments. We have collected a diverse range of dental aligment, we are not solely collecting cases with excessive overbites or perfectly aligned teeth. This model can reliably determine the presence or absence of crowding, maxillary protrusion, and deep overbite even in cases presenting with open bite or anterior crossbite. As mentioned earlier, for each case, we comprehensively assessed various malocclusions (including open bite, crossbite, and mandibular protrusion etc.). For each subject, we annotated the presence or absence of multiple types of malocclusion. For example, for the maxillary protrusion dataset, we constructed the dataset by sorting it into 0/1 based on whether maxillary protrusion was present or absent. This means that since other malocclusion types were also assessed, a person labeled 0 for maxillary protrusion does not necessarily mean they have no other malocclusions, and a person labeled 1 does not mean they only have maxillary protrusion. Thus, both the 0 and 1 sides of the constructed dataset are rich in diverse dental alignment. Since the learning model was built on this foundation, it can detect other malocclusions even when present, making this classification algorithm versatile.

This study demonstrates the potential and usefulness of algorithms utilizing AI and smartphone images; it does not represent a finished product ready for implementation. We currently plan to gather more cases to build classification models for each malocclusion type( include open bite, crossbite, etc.). By constructing binary classification models for all types of dental malocclusion and connecting these multiple algorithms in parallel, it will be possible to assess the risk for each malocclusion type from a single data input. This will enable us to implement a screening tool for risk assessment across all possible dental malocclusion.

5. Presentation of Demographic Data:

On pages 21–22, the section describing “Residential area” and “Annual household income” is written in incomplete sentences and lacks c

---

## [Decision Letter · Decision Letter 3]

27 Nov 2025

Accuracy of AI-based binary classification for detecting malocclusion in the mixed dentition stage

PONE-D-24-52219R3

Dear Dr. Oka,

We’re pleased to inform you that your manuscript has been judged scientifically suitable for publication and will be formally accepted for publication once it meets all outstanding technical requirements.

Kind regards,

Claudia Trindade Mattos, Ph.D.

Academic Editor

PLOS ONE

Additional Editor Comments (optional):

All comments have been addressed and the manuscript now meets the standards for publication.

Reviewers' comments:

Reviewer's Responses to Questions

**Comments to the Author**

Reviewer #4: All comments have been addressed

2. Is the manuscript technically sound, and do the data support the conclusions?

Reviewer #4: Yes

3. Has the statistical analysis been performed appropriately and rigorously?

Reviewer #4: Yes

4. Have the authors made all data underlying the findings in their manuscript fully available?

Reviewer #4: No

5. Is the manuscript presented in an intelligible fashion and written in standard English?

Reviewer #4: Yes

Reviewer #4: Thank you for addressing my previous comments. I am glad to see an improved version of this manuscript.

**Do you want your identity to be public for this peer review?** For information about this choice, including consent withdrawal, please see our Privacy Policy

Reviewer #4: No

---

## [Editor Report · Acceptance letter]

PONE-D-24-52219R3

PLOS ONE

Dear Dr. Oka,

I'm pleased to inform you that your manuscript has been deemed suitable for publication in PLOS ONE. Congratulations! Your manuscript is now being handed over to our production team.

Kind regards,

on behalf of

Dr. Claudia Trindade Mattos

Academic Editor

PLOS ONE